# Importance of broken geometric symmetry of single-atom Pt sites for efficient electrocatalysis

Junsic Cho [1,10], Taejung Lim [2,10], Haesol Kim[1,10], Ling Meng[3], Jinjong Kim [2,4], Seunghoon Lee[1], Jong Hoon Lee[5], Gwan Yeong Jung[6], Kug-Seung Lee [7], Francesc Viñes [3], Francesc Illas [3], Kai S. Exner[8] ✉, Sang Hoon Joo [2,4] ✉ & Chang Hyuck Choi [1,9] ✉

Platinum single-atom catalysts hold promise as a new frontier in heterogeneous electrocatalysis. However, the exact chemical nature of active Pt sites is highly elusive, arousing many hypotheses to compensate for the significant discrepancies between experiments and theories. Here, we identify the stabilization of low-coordinated $Pt^{II}$ species on carbon-based Pt single-atom catalysts, which have rarely been found as reaction intermediates of homogeneous $Pt^{II}$ catalysts but have often been proposed as catalytic sites for Pt single-atom catalysts from theory. Advanced online spectroscopic studies reveal multiple identities of $Pt^{II}$ moieties on the single-atom catalysts beyond ideally four-coordinated $Pt^{II}–N_4$. Notably, decreasing Pt content to 0.15 wt.% enables the differentiation of low-coordinated $Pt^{II}$ species from the four-coordinated ones, demonstrating their critical role in the chlorine evolution reaction. This study may afford general guidelines for achieving a high electrocatalytic performance of carbon-based single-atom catalysts based on other $d^8$ metal ions.

Downsizing bulk metal catalysts to the atomic level, alias single-atom catalysts (SACs), is a promising strategy for realizing superior catalytic activity and reducing costs[1–5]. SACs are often considered bridges between heterogeneous and homogeneous catalysis, and ideally, they bring hope to realize a uniform active site distribution, as encountered in homogeneous catalysis, in the realm of heterogeneous catalysts[4,5]. However, increasing literature suggests that SACs exhibit a significant degree of heterogeneity in their active sites[6–8]. Consequently, this

heterogeneity hinders the identification of the genuine active site and the precise quantification of the intrinsic activity of the catalysts by turnover frequency (TOF) or similar metrics, thus causing formidable challenges for the rational design of better SACs.

In this work, we focus on archetypal Pt SACs, which have found broad utility in a variety of electro-, photo-, and heterogeneous catalytic reactions[9–12]. In general, Pt SACs are composed of Pt ions in an oxidation state of $+2$[11–14]. Because $Pt^{II}$ has a $d^8$ electronic configuration,

[1]Department of Chemistry, Pohang University of Science and Technology (POSTECH), Pohang 37673, Republic of Korea. [2]Department of Chemistry, Ulsan National Institute of Science and Technology (UNIST), Ulsan 44919, Republic of Korea. [3]Departament de Ciència de Materials i Química Física & Institut de Química Teòrica i Computacional (IQTCUB), Universitat de Barcelona, c/ Martí i Franquès 1-11, 08028 Barcelona, Spain. [4]Department of Chemistry, Seoul National University, Seoul 08826, Republic of Korea. [5]UNIST Central Research Facilities (UCRF), Ulsan National Institute of Science and Technology (UNIST), Ulsan 44919, Republic of Korea. [6]School of Energy and Chemical Engineering, Ulsan National Institute of Science and Technology (UNIST), Ulsan 44919, Republic of Korea. [7]Beamline Department, Pohang Accelerator Laboratory, Pohang University of Science and Technology (POSTECH), Pohang 37673, Republic of Korea. [8]Faculty of Chemistry, Theoretical Inorganic Chemistry, University of Duisburg-Essen, 45141 Essen, Germany; Cluster of Excellence RESOLV, 44801 Bochum, Germany; Center for Nanointegration Duisburg-Essen (CENIDE), 47057 Duisburg, Germany. [9]Institute for Convergence Research and Education in Advanced Technology (I-CREATE), Yonsei University, Seoul 03722, Republic of Korea. [10]These authors contributed equally: Junsic Cho, Taejung Lim, Haesol Kim. ✉e-mail: kai.exner@uni-due.de; shjoo1@snu.ac.kr; chchoi@postech.ac.kr

it strongly prefers to be stabilized by a near-perfect square planar geometry ($D_{4h}$), as expected from the ligand field theory[15]. Thus, based on conventional extended X-ray absorption fine structure (EXAFS) analyses, the coordination environments of $Pt^{II}$ in carbon-supported Pt SACs are typically accepted as a porphyrin-like geometry, such as $Pt-N_4$ and $Pt-S_4$, where p-block elements doped in carbon substrates function as surface pockets for immobilizing Pt ions[16–19]. However, in theory, these model structures predict substantially weakened axial coordination and, consequently, poor catalytic activity, while broken or unsaturated ones are expected to offer more optimized axial coordination leading to high catalytic activity[12,20–23]. These phenomena originate from a significant ligand field splitting ($\Delta$), which effectively upshifts the energy level of the empty $d_{x^2-y^2}$ orbital, but lowers that of the fully occupied other d orbitals. Thus, a considerable energy cost is required for the charge redistribution of $Pt^{II}$ for its strong axial bond formation. Therefore, discrepancies between the experimentally observed electrocatalytic performance of carbon-supported Pt SACs and theoretically computed models have been reported in the literature[24].

Extensive efforts have thus been made to identify the chemical nature of the catalytic sites in Pt SACs. Possible asymmetric coordination geometries such as $Pt-N_xC_{4-x}$ have been proposed by density functional theory (DFT) calculations[19,25,26]. Additionally, in situ modifications of the symmetric coordination geometries, formed by partial ligand exchange of the four equivalent ligands with electrolyte components (e.g., water) or by changes in the oxidation state of the central $Pt^{II}$ ions under realistic electrochemical conditions, have also been considered[12,19,27]. Despite previous achievements, which indicate the poor or moderate catalytic performance of Pt sites with symmetric $D_{4h}$ geometry, the broken geometric symmetry of Pt SACs and their key roles in electrocatalysis have not been clarified thus far. This uncertainty primarily originates from the possible heterogeneity of isolated Pt sites, considering that not all sites are necessarily equally active toward the investigated electrochemical reaction. Unfortunately, even EXAFS analysis, which is likely the most powerful and extensively used technique for studying the structure of a single site, cannot conclusively distinguish between different Pt moieties that coexist in SACs. Thus, the synthesis of SACs containing homogeneous catalyst-like single active sites is crucial for the clear identification of the genuine active sites in Pt SACs.

This study reveals the presence of low-coordinated $Pt^{II}$ species in carbon-based Pt SACs and their vital roles in electrocatalysis. The chlorine evolution reaction (CER) is taken as a model reaction, of which the product, $Cl_2$, is practically important due to extensive applications in the chemical industry, typically produced via the chlor-alkali process with Ru/Ir-based dimensionally stable anode (DSA) electrodes[28,29]. We quantify the extent of Pt demetallation from Pt SACs in operando using advanced online inductively coupled plasma-mass spectrometry coupled with an electrochemical flow cell (EFC/ICP-MS) and compare the result with the simultaneous electrochemical activity loss during the CER. No singular Pt identity is confirmed from the considerable discrepancy between these two parameters, i.e., Pt dissolution vs. CER activity loss, without significant modification of the TOF. In addition to the common symmetric $D_{4h}$ geometry, low-coordinated $Pt^{II}$ species is clearly observed in the EXAFS spectrum of a control Pt SAC with an ultralow Pt loading of 0.15 wt.%. Structure-dependent studies of the catalytic activity, selectivity, and stability by combining experimental and theoretical approaches verify the central role of the low-coordinated $Pt^{II}$ sites in Pt SACs for boosting the CER.

## Results

A model catalyst was prepared by heat treatment of a powder mixture of $Pt^{II}$ meso-tetraphenylporphine (PtTPP) and acid-treated carbon nanotubes (CNTs) at 700 °C under $N_2$ flow. The prepared catalyst was characterized as in our previous studies using high-angle annular dark-

field scanning transmission electron microscopy (HAADF-STEM), X-ray diffraction (XRD), X-ray absorption spectroscopy (XAS), and X-ray photoelectron spectroscopy (XPS)[17,30]. The catalyst is composed of abundant porphyrin-like $Pt^{II}-N_4$ moieties, which are covalently embedded on the CNT support (cf. Supplementary Note 1 for a detailed discussion; Supplementary Figs. 1–5). The Pt content is approximately 3 wt.%, as determined by inductively coupled plasma-optical emission spectroscopy (ICP-OES). This model catalyst is hereafter named 'Pt$_1$(3)/CNT', where the subscript '1' indicates the atomic isolation of Pt, and the number in parenthesis indicates the Pt content in wt.%.

The catalytic performance of Pt$_1$(3)/CNT was evaluated for the CER, the anodic reaction of chlor-alkali electrolysis to produce gaseous chlorine by a two-electron process — $2Cl^- \rightarrow Cl_2 + 2e^-$, $U^0_{CER} = 1.36$ V vs. standard hydrogen electrode (SHE)[31,32] — that competes with the oxygen evolution reaction (OER; $2H_2O \rightarrow O_2 + 4H^+ + 4e^-$, $U^0_{OER} = 1.23$ V vs. reversible hydrogen electrode (RHE)). Pt$_1$(3)/CNT shows an excellent CER activity (Fig. 1a). While no significant Faradaic current is observed in a NaCl-free 0.1 M $HClO_4$ electrolyte, with the addition of 1 M NaCl into the 0.1 M $HClO_4$ electrolyte, the catalyst records an onset potential of 1.36 $V_{RHE}$ and an oxidation current density ($j$) of 53 mA cm$^{-2}$ at 1.45 $V_{RHE}$. The online differential electrochemical mass spectrometry (DEMS) measurement reveals a predominant ionic current for $m/z = 35$ (Cl$^+$) in 0.1 M $HClO_4$ + 1 M NaCl electrolyte (Fig. 1b), which corresponds to the fragmentation of $Cl_2$ and its hydrolyzed derivatives from the following equation: $Cl_2 + H_2O \rightarrow HCl + HOCl$[32,33]. Concomitantly, distinct ionic currents for $m/z = 36$ and 51 (HCl$^+$ and OCl$^+$, respectively) are detected, indicating the formation of HCl and HOCl during CER. An insignificant ionic current for $m/z = 32$ (O$_2^+$) may attribute to the further decomposition of HOCl forming $O_2$ via the following equation: $2HOCl \rightarrow 2HCl + O_2$ (cf. Supplementary Note 2 for a detailed discussion of non-Faradaic $O_2$ formation)[32,33]. In NaCl-free 0.1 M $HClO_4$ electrolyte, the $O_2$ is not detected within the CER-relevant potential window, inferring that the Pt$_1$(3)/CNT catalyzes CER selectively against competitive OER (Fig. 1c). The DEMS results are further corroborated by the rotating ring disk electrode (RRDE) measurement, which exhibits approximately 100% CER selectivity (Supplementary Fig. 6). Notably, this CER activity outperforms that of the commercial DSA (Supplementary Fig. 7) and most of the previously reported CER catalysts (Supplementary Table 1)[34–39].

The high catalytic activity of Pt$_1$(3)/CNT is primarily attributed to the presence of Pt sites. CNT and N-doped CNT — the latter was synthesized with a Pt-free TPP precursor — exhibit much smaller oxidation currents than Pt$_1$(3)/CNT (Fig. 1a). In addition, the CER activity of Pt$_1$(3)/CNT deteriorates considerably in a CO-saturated electrolyte (Fig. 1d), a well-known poisoning agent with a high binding affinity for atomically isolated $Pt^{II}$ ions[27,40]. This result further confirms the catalytic role of Pt sites in the efficient CER. The polarization curves measured in the Ar- or CO-saturated electrolytes without NaCl are almost identical, indicating that the decrease in CER activity is not an artifact induced by the competitive CO oxidation reaction under anodic polarization conditions.

After confirming the high CER activity of Pt$_1$(3)/CNT and the chemical nature of its catalytic site, the durability of this catalyst was evaluated by measuring 500 iterative cyclic voltammograms (CVs) from 1.0 to 1.6 $V_{RHE}$ (Fig. 1e). We adopted iterative CVs for accelerating catalyst degradation, as this catalyst exhibits promising stability under constant potential conditions relevant to real CER electrolysis conditions (Supplementary Fig. 8). After 500 CVs, Pt$_1$(3)/CNT exhibits a significant decrease in CER activity, and the $j$ value measured at 1.45 $V_{RHE}$ decreases by 61% from 53 to 21 mA cm$^{-2}$ (Fig. 1f). The CER activity decreases rapidly at the beginning of the durability test, but then it becomes gradually alleviated.

To reveal the fundamental origin of the observed deactivation, real-time Pt dissolution from Pt$_1$(3)/CNT was analyzed using online

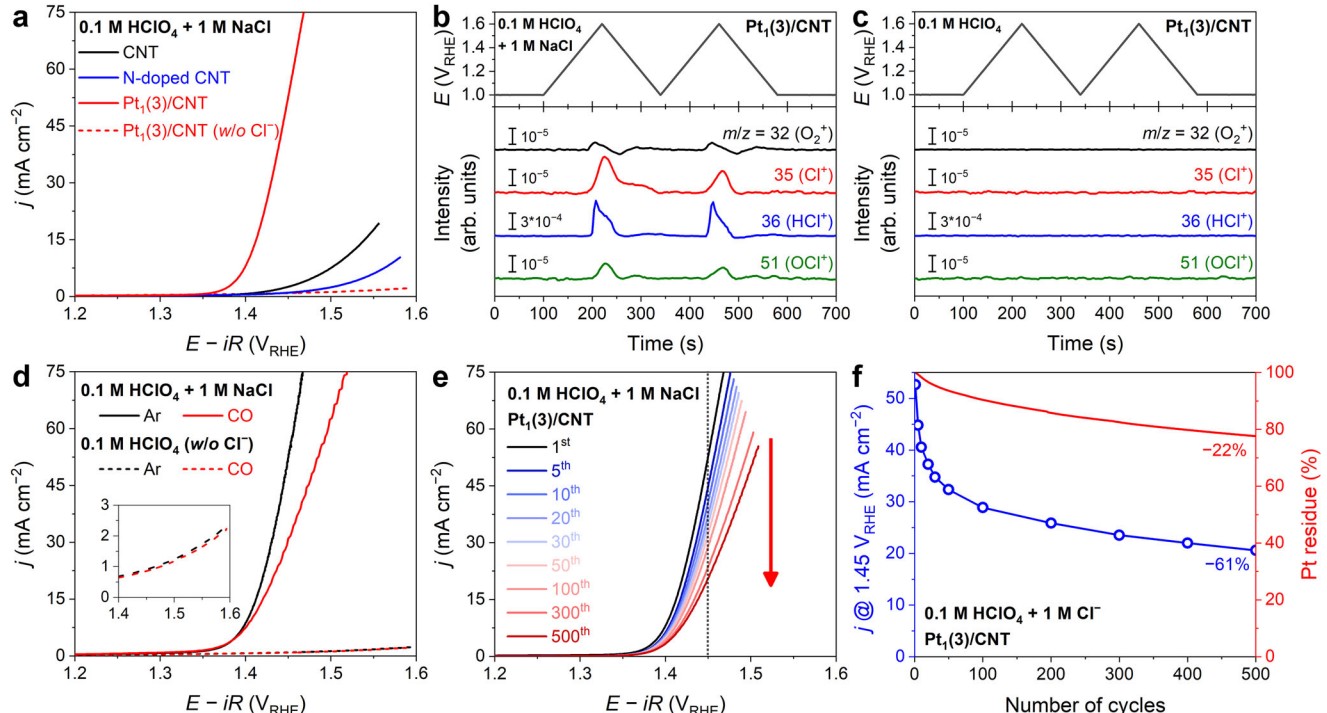

**Fig. 1 | CER performance of Pt$_1$(3)/CNT. a** CER polarization curves of the CNT, N-doped CNT, and Pt$_1$(3)/CNT catalysts obtained in Ar-saturated 0.1 M HClO$_4$ with 1 M NaCl. The polarization curve of Pt$_1$(3)/CNT measured in an NaCl-free electrolyte is also shown (dotted line). **b, c** Online DEMS results of $m/z$ = 32, 35, 36, and 51 of Pt$_1$(3)/CNT during two consecutive slow CVs obtained in Ar-saturated 0.1 M HClO$_4$ with 1 M NaCl (**b**) and NaCl-free electrolytes (**c**). **d** The polarization curves of Pt$_1$(3)/CNT in Ar/CO-saturated 0.1 M HClO$_4$ with and without 1 M NaCl. **e** The durability test of Pt$_1$(3)/CNT performed by measuring CER polarization curves during iterative 500 CVs from 1.0 and 1.6 V$_{RHE}$. **f** Comparison between the CER activity decrement and Pt loss measured during the durability test.

EFC/ICP-MS in an Ar-saturated 0.1 M HClO$_4$ electrolyte containing 1 M NH$_4$Cl (Supplementary Figs. 9 and 10). By applying the same CV conditions as those used for the durability study, the Pt sites remaining on the Pt$_1$(3)/CNT were estimated by subtracting the accumulated amount of dissolved Pt ions from the initial Pt content (Supplementary Fig. 11). The total Pt loss after 500 CVs is only 22% (Fig. 1f), which hardly corresponds to the considerable CER activity loss of 61%. Indeed, the Pt dissolution profile reveals relatively no significant Pt demetallation at the beginning of these experiments.

Consequently, we considered a possible TOF modification of the aged Pt$_1$(3)/CNT along with Pt dissolution. It is important to note that the apparent catalytic activity is not only governed by the active site density but also by the TOF of each catalytic site[41–43]. TOF modification of SACs typically originates from structural changes in the local coordination geometry or the introduction of new heteroatoms or functional groups near their catalytic metal sites[20,44,45]. However, the $k^3$-weighted Pt L$_3$-edge EXAFS spectrum measured after 500 CVs reveals an almost identical Pt–N bond length and Pt–N coordination number (CN) to those of the pristine Pt$_1$(3)/CNT (Supplementary Fig. 12), indicating no significant structural change in the active Pt sites after the durability test.

On the other hand, the XPS spectrum pinpoints substantial alternations in the chemical composition of Pt$_1$(3)/CNT after 500 CVs. The XPS O 1$s$ spectrum shows a new peak at 532.3 eV, indicating the formation of C=O (531.5 eV) and C–O (532.6 eV) functionalities on Pt$_1$(3)/CNT after the durability test (Fig. 2a and Supplementary Fig. 13)[46]. Also, the aged Pt$_1$(3)/CNT shows a redox couple at 0.56 V$_{RHE}$, which is a fingerprint of oxygen functional groups (e.g., quinone) on the carbon surface[47,48], and this peak intensifies as the number of CV cycles increases (Fig. 2b). In addition to oxygen functionalities, chlorine functional groups are also formed. The XPS Cl 2$p$ spectrum of the aged Pt$_1$(3)/CNT reveals peaks at 200.4 eV with a spin-orbit splitting of

1.6 eV (Fig. 2c). This peak corresponds to the core level spectra of organochlorine compounds, not residual alkali chloride (199 eV)[49], indicating the cogeneration of chlorine functional groups on the carbon support after 500 CVs.

To identify whether the newly generated functional groups are responsible for the decrease in the TOF of Pt$_1$(3)/CNT, two model catalysts with abundant oxygen (O-Pt$_1$(3)/CNT) and chlorine (Cl-Pt$_1$(3)/CNT) functionalities were additionally prepared by post-treatment of Pt$_1$(3)/CNT with O$_3$ and SO$_2$Cl$_2$, respectively[50]. ICP-OES analysis revealed no detectable Pt loss after these post-treatments. In contrast, XPS O 1$s$/Cl 2$p$ spectra (and electrochemical redox signals at 0.56 V$_{RHE}$ for O-Pt$_1$(3)/CNT) confirm the successful introduction of the oxygen and chlorine functional groups onto the CNT support (Fig. 2a–c). Despite the introduction of the functional groups, the CER polarization curves for O-Pt$_1$(3)/CNT and Cl-Pt$_1$(3)/CNT are almost identical to that of pristine Pt$_1$(3)/CNT (Fig. 2d). These results indicate that the catalytic degradation of aged Pt$_1$(3)/CNT is likely not attributable to the TOF modification induced by the newly generated oxygen or chlorine functional groups.

In addition to the functional group generation, the other considerable change in Pt$_1$(3)/CNT after 500 CVs was the decrease in the Pt content of the catalysts (Fig. 1f). Although Pt moieties are the catalytic sites for the CER, their electronic structures can be affected by their content. Since the Pt moieties are implanted on the CNT surface, their conjugation can disturb the electronic structure of the carbon support (e.g., electron withdrawing/donating properties) and consequently tune the TOF of the Pt sites[51]. Besides the carbon support modifications, a recent study on Fe SACs also suggested direct electronic interactions among adjacent Fe moieties, namely TOF modifications, induced by the spatial proximity of the Fe sites as the Fe loading increased[52]. Therefore, a decrease in the Pt active site density may be responsible for the TOF decay, possibly leading to a rapid decrease in the CER activity.

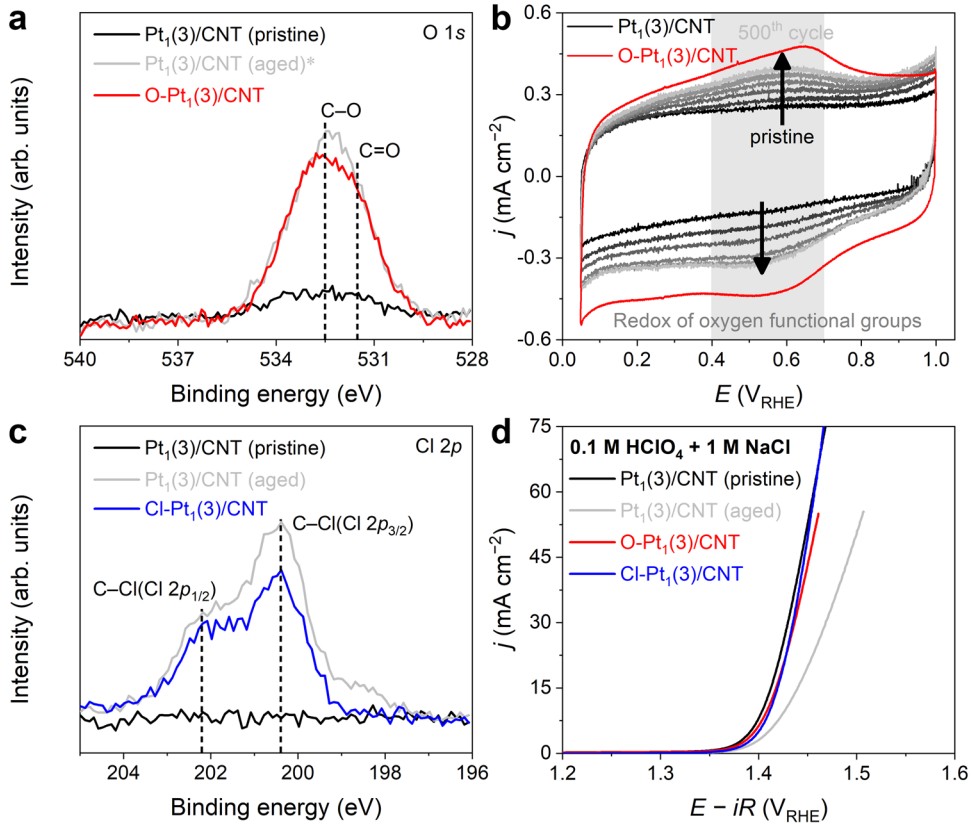

**Fig. 2 | In/ex situ surface modifications of Pt₁(3)/CNT. a** The XPS O 1s spectra of Pt₁(3)/CNT, aged Pt₁(3)/CNT, and O-Pt₁(3)/CNT. (*Nafion contribution-subtracted spectrum; See Supplementary Fig. 13). **b** CV responses of pristine and aged Pt₁(3)/CNT measured in Ar-saturated 0.1 M HClO₄. The CV response of O-Pt₁(3)/CNT is also

shown. **c** The XPS Cl 2p spectra of Pt₁(3)/CNT, aged Pt₁(3)/CNT, and Cl-Pt₁(3)/CNT. **d** CER polarization curves of the Pt₁(3)/CNT, aged Pt₁(3)/CNT, O-Pt₁(3)/CNT, and Cl-Pt₁(3)/CNT catalysts obtained in Ar-saturated 0.1 M HClO₄ with 1 M NaCl.

Based on this deduction, Pt₁/CNTs were synthesized with different Pt loadings of 1 and 0.15 wt.%, which were denoted as 'Pt₁(1)/CNT' and 'Pt₁(0.15)/CNT', respectively. The structures of the two catalysts were extensively characterized (Supplementary Figs. 1–5), showing the formation of Pt SACs as Pt₁(3)/CNT. Interestingly, despite the substantial decrease in Pt loading, Pt₁(1)/CNT and Pt₁(0.15)/CNT exhibit only a slight decrease in CER activity compared with Pt₁(3)/CNT (Fig. 3a). The polarization curves show $j$ values of 53, 49, and 39 mA cm⁻² at 1.45 $V_{RHE}$ as the Pt content decreases from 3 wt.% to 1 and 0.15 wt.%, respectively. Independent of Pt loading, the CER selectivity is almost 100% (Supplementary Figs. 6 and 14). Notably, the Pt-loading-dependent CER activity can be translated into three and fifteen times higher TOF values for Pt₁(1)/CNT and Pt₁(0.15)/CNT, respectively, compared with Pt₁(3)/CNT. The observed TOF trend contradicts our deduction, which was that the TOF decreases with decreasing Pt content. Thus, the control experiments suggest that the TOF decay, induced either by newly generated oxygen-/chlorine-functional groups or the loss of Pt moieties, is not the main cause of Pt₁/CNT deactivation.

Therefore, we searched for the origin of the unexpected TOF trend, and the XAS spectra of Pt₁(1)/CNT and Pt₁(0.15)/CNT provided a decisive clue (Fig. 3b). Similar to Pt₁(3)/CNT, the X-ray absorption near edge structure (XANES) spectra of Pt₁(1)/CNT and Pt₁(0.15)/CNT show the +2 oxidation state of Pt (Supplementary Fig. 4). Their EXAFS spectra also specify a strong Pt–N scattering at 2.0 Å without Pt–Pt scattering at 2.8 Å (Supplementary Fig. 3 and Supplementary Table 2). Interestingly, the Pt–N peak intensity decreases as the Pt loading decreases, indicating the different coordination natures of the Pt sites in the three control catalysts. Their fitting parameters show lowered CN values from 4.0 for Pt₁(3)/CNT to 3.4 for Pt₁(1)/CNT and further to 3.0 for Pt₁(0.15)/CNT (Fig. 3c and Supplementary Table 2). Considering

that the $d^8$ configuration of Pt^II prefers a four-coordinated square planar structure (e.g., Pt^II–N₄)[15], the CN value of 3.0 indicates an unusual stabilization of isolated Pt^II in the form of either trigonal-planar-like Pt^II–N₃ or T-shaped Pt^II–N₃V (where V denotes a vacancy). Notably, these coordinatively unsaturated Pt^II complexes are very rarely found so far but have often been proposed as key reaction intermediates in homogeneous catalysis[53]. Hence, the coexistence of the coordinatively unsaturated Pt–N₃(V) sites (i.e., Pt–N₃ and/or Pt–N₃V) with Pt–N₄ on the heterogeneous CNT support emphasizes the failure of all earlier discussions to understand the fundamental origin of Pt₁(3)/CNT deactivation because the assumption of a singular catalytic site corresponding to Pt–N₄ is violated.

Notably, the considerable CER activity of Pt₁(0.15)/CNT demonstrates that Pt–N₃(V) rather than the symmetric Pt–N₄ site is responsible for the excellent CER activity. Thus, the apparent CER activity of Pt₁/CNT catalysts depends neither on the total Pt content nor on the total number of Pt–N₄ sites but is mainly governed by the number of Pt–N₃(V) sites. Therefore, we approximately predicted the Pt–N₃(V) contents of Pt₁(1)/CNT and Pt₁(3)/CNT by linear extrapolation to the corresponding $j$ values of an extended line defined from zero current with no Pt–N₃(V) content to $j$ value and Pt–N₃(V) content of Pt₁(0.15)/CNT (Fig. 3d). By subtracting the Pt–N₃(V) content from the total Pt content, the Pt–N₄ contents of Pt₁(1)/CNT and Pt₁(3)/CNT can also be derived. The calculation determines the Pt–N₃(V)/Pt–N₄ contents to be 0.19/0.81 and 0.2/2.8 wt.% for Pt₁(1)/CNT and Pt₁(3)/CNT, respectively. Based on these approximations, which are only valid when the TOFs of the active Pt–N₃(V) sites for all Pt₁/CNT catalysts are identical, the average CN values can be inversely estimated to be 3.8 and 3.9 for Pt₁(1)/CNT and Pt₁(3)/CNT, respectively. These values are within the error range of the EXAFS fitting parameters (Fig. 3c).

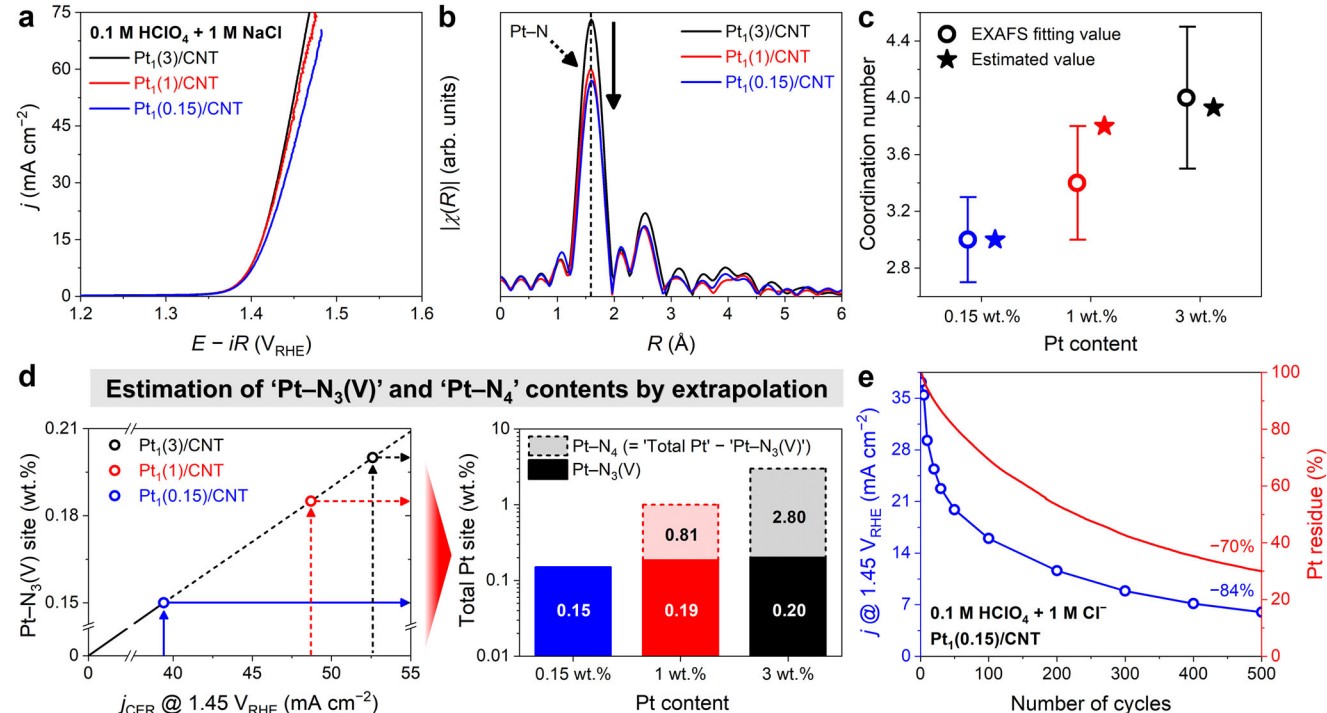

**Fig. 3 | Pt loading effects on CER activity of Pt$_1$/CNTs. a** CER polarization curves of the Pt$_1$(3)/CNT, Pt$_1$(1)/CNT, and Pt$_1$(0.15)/CNT catalysts obtained in Ar-saturated 0.1 M HClO$_4$ with 1 M NaCl. **b** The $k^3$-weighted Pt L$_3$-edge EXAFS spectra of the Pt$_1$/CNT catalysts without phase correction and **c** their Pt–N coordination numbers obtained by EXAFS fitting. The error bars indicate the error ranges of the EXAFS fitting parameter. **d** Estimation of Pt-N$_3$(V) contents in Pt$_1$(1)/CNT and Pt$_1$(3)/CNT by linear extrapolation to their corresponding $j$ values of an extended line defined from zero current with no Pt-N$_3$(V) content to $j$ value and Pt-N$_3$(V) content of Pt$_1$(0.15)/CNT. **e** Comparison between the CER activity decrement and Pt loss of Pt$_1$(0.15)/CNT during the durability test.

In addition, the critical role of Pt-N$_3$(V) in catalyzing CER is further corroborated by in situ XAS measurements. In the Pt L$_3$-edge XANES spectra, which were measured in an Ar-saturated 0.1 M HClO$_4$ + 1 M NaCl electrolyte, the white line (WL) intensity marginally increases after immersing the catalyst into the electrolyte, and the increment further intensifies at 1.45 V$_{RHE}$ (Supplementary Fig. 15). This result agrees well with our previous study and infers the adsorption of CER intermediates on the Pt sites[30]. Interestingly, the intensified WL at 1.45 V$_{RHE}$ is slightly higher for Pt$_1$(0.15)/CNT and decreases with increasing Pt content in the catalysts, indicating higher coverage of CER intermediates as a proportion of Pt-N$_3$(V) sites in the catalysts increases. The same conclusion is also made with in situ EXAFS spectra that show an increasing Pt-Cl scattering peak at 2.3 Å as Pt content in the catalysts decreases (Supplementary Fig. 15 and Supplementary Table 3). We further note that CN$_{Pt-N}$ of Pt$_1$(0.15)/CNT increases from 3 for the powdery sample (Fig. 3c) to 4 under the electrochemical conditions and attribute this change to in situ formation of an additional Pt-O bond, which will be discussed again in the DFT section. The ratio between CN$_{Pt-N/O}$ and CN$_{Pt-Cl}$ of Pt$_1$(0.15)/CNT is approximately 4 : 1, and this supports the predominant presence of Pt-N$_3$(V) sites in Pt$_1$(0.15)/CNT and their high catalytic activity towards CER.

The identification of the main catalytic site leads to a much-alleviated disparity between the extent of CER activity drop and accumulated Pt loss, as observed in the example of Pt$_1$(0.15)/CNT during its durability test (Fig. 3e). After 500 CVs, the initial CER activity and Pt content of Pt$_1$(0.15)/CNT decrease by 84 and 70%, respectively, and their profiles over time are also comparable. In addition, the Pt loss of Pt$_1$(X)/CNTs (X = 0.15, 1, and 3) after 500 CVs becomes intensified as the proportion of the Pt-N$_3$(V) increases (Supplementary Fig. 16). These results further corroborate that the Pt-N$_3$(V) site is the genuine catalytic site and further indicate that the decrease in CER activity after the durability test primarily originates from the loss of Pt-N$_3$(V) sites.

Therefore, for Pt$_1$(3)/CNT, the significant discrepancy between the activity drop (−61%) and Pt loss (−22%) can now be accounted by the coexistence of the Pt-N$_3$(V) minority and Pt-N$_4$ majority moieties; the former site is more active but also more labile than the latter.

To comprehend the detailed CER path on the Pt$_1$/CNT catalysts and their catalytic structure, we apply electronic structure calculations in the framework of DFT. Based on the EXAFS fitting parameters, we consider three different models, namely square planar Pt-N$_4$, trigonal planar Pt-N$_3$, and T-shaped Pt-N$_3$V (Fig. 4a). For all these models, the structures of the catalytically active Pt site were characterized under CER conditions ($U > 1.36$ V$_{SHE}$) by the construction of Pourbaix diagrams[54]. While an axially unoccupied Pt site (*) is favored for Pt-N$_4$ (Supplementary Figs. 17 and 18), Pt-N$_3$ and Pt-N$_3$V are capped by oxygen, *O, which, together with the Pt atom underneath, serves as the active site (Supplementary Fig. 18). These results suggest that the CER over Pt-N$_4$ and Pt-N$_3$(V) proceeds via *Cl or *OCl intermediates, respectively. We have modeled both pathways for all three sites and quantified the largest free-energy span between the intermediate states in dependence on applied electrode potential by referring to the descriptor $G_{max}(U)$[55]. The compilation of the free-energy diagrams at $U = 1.36$ V$_{SHE}$ reveals that, in agreement with our previous works[17,30,56], square planar Pt-N$_4$ prefers the *Cl ($G_{max}(U) = 0.32$ eV) rather than the *OCl path ($G_{max}(U) = 1.07$ eV) (Fig. 4b). However, the *OCl path is energetically favored over the *Cl mechanism for Pt-N$_3$ or Pt-N$_3$V (Fig. 4c, d), inferring the different CER mechanisms for the Pt-N$_4$ and Pt-N$_3$(V) moieties. Notably, the activity descriptor $G_{max}(U)$ amounts to 0.21 and 0.05 eV for Pt-N$_3$ and Pt-N$_3$V at $U = 1.36$ V$_{SHE}$, respectively, confirming the experimental result that the Pt-N$_3$(V) sites are more active than Pt-N$_4$ in the CER. In addition, the competing OER for the three different sites was described by assuming the well-accepted mononuclear mechanism via the *OH, *O, and *OOH adsorbates, and $G_{max}(U)$ as a measure for the electrocatalytic activity was determined (Supplementary Fig. 19 and Supplementary Table 4)[57].

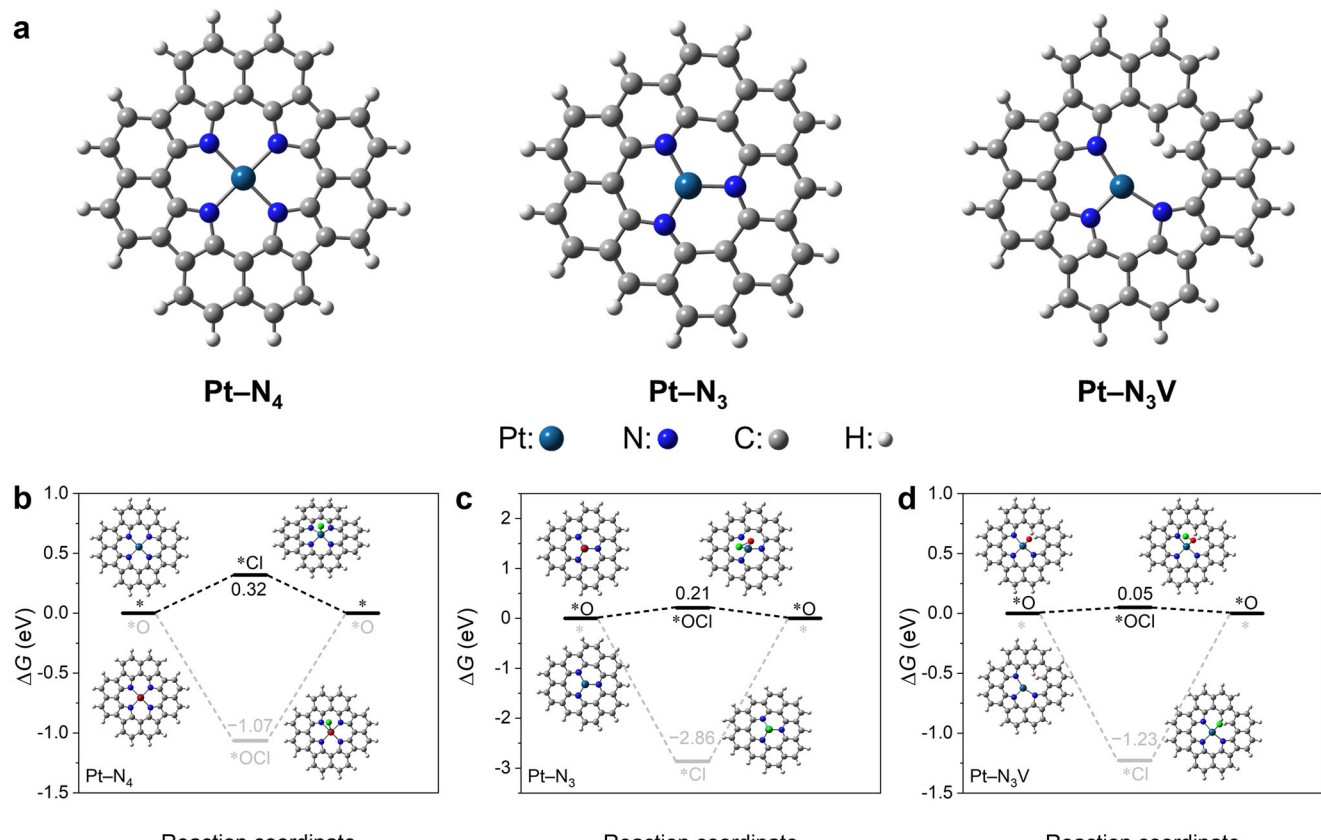

**Fig. 4 | Identification of the catalytic sites by DFT calculations. a** Three active site models, square planar Pt–N$_4$, trigonal planar Pt–N$_3$, and T-shaped Pt–N$_3$V, used for the DFT calculations. **b**–**d** Free-energy diagrams of the CER via the *Cl and *OCl pathways at 1.36 V$_{SHE}$ for Pt–N$_4$ (**b**), Pt–N$_3$ (**c**), and Pt–N$_3$V (**d**).

By applying a selectivity model for the competing CER and OER[58], we demonstrate that CER selectivity amounts to 100% (Supplementary Fig. 20), independent of the applied electrode potential or chemical nature of the active sites, which coincides with the experimental results (Fig. 1b and Supplementary Figs. 6 and 14).

Finally, we study the stability of the Pt–N$_4$ and Pt–N$_3$(V) moieties under the CER conditions. For this purpose, the equilibrium potential ($U_{diss}$) for the oxidative demetallation of central Pt species to PtO$_2$ is introduced as a catalyst stability descriptor (cf. DFT calculation section in Methods and Supplementary Note 3). It is of note that PtO$_2$ is the energetically favored phase under CER conditions, according to the Pourbaix diagram[59]. The results show $U_{diss}$ of 2.98, 1.99, and 0.78 V$_{SHE}$ for Pt–N$_4$, Pt–N$_3$V, and Pt–N$_3$, respectively. Consistent with the experimental results, the descriptor $U_{diss}$ indicates the poorer stability of Pt–N$_3$(V) compared with Pt–N$_4$ under CER conditions of $U > 1.36$ V$_{SHE}$. Considering the $U_{diss}$ for Pt–N$_3$ amounts to 0.78 V$_{SHE}$, which is significantly below the CER equilibrium potential, we conclude that the trigonal planar Pt–N$_3$ site cannot be stable under the CER conditions, but instead, T-shaped Pt–N$_3$V is responsible for the excellent CER activity and selectivity of the Pt$_1$/CNT catalysts.

In this work, we have unraveled the fundamental origin of the high CER activity of Pt$_1$/CNT catalysts. The heterogeneity of central Pt$^{II}$ ions and the key role of three-coordinated Pt$^{II}$ with broken $D_{4h}$ symmetry in CER electrocatalysis were demonstrated. Thus, the development of new synthetic strategies to maximize the site density of three-coordinated Pt with a broken symmetric geometry will become an upcoming challenge. This originates from the limited amount of Pt–N$_3$(V) on the catalyst surface, which apparently converges to 0.2 wt.% only, regardless of the increasing total Pt content on Pt$_1$/CNT. Indeed, tailoring the coordination geometry will also be critical to find optimal ligand-field strength for stabilizing the Pt sites and *OCl intermediate, owing to the trade-off

relation between stability and activity. Otherwise, more practical approaches, such as engineering the catalyst-electrolyte interface or finding optimal operating conditions (e.g., well-regulated potential excursions) to promote the longevity of labile Pt–N$_3$(V) sites, will be the subject of another future research direction. Before answering these questions, however, understanding the underlying fundamental origins of the (quasi-)stabilization of the rarely found Pt–N$_3$(V) species on the supporting substrates should be prioritized. Considering the cases of Ni$^{II}$ SACs for electrochemical CO$_2$ reduction[20], these guidelines might be general tasks for other metal ions with a $d^8$ electronic configuration in SACs, not limited to Pt$^{II}$ catalytic sites. Therefore, the present findings will provide a foundation for the rational design of next-generation SACs with excellent electrocatalytic performances.

## Methods

### Preparation of the Pt$_1$/CNT catalysts

Before synthesizing the catalysts, multiwalled CNTs (MR99, Carbon Nano-material Technology Co., Ltd.) with an average diameter of 10 nm and an average length of 10 μm were heated and subsequently acid washed to remove metallic impurities[60]. In detail, CNTs (38.0 g) were calcined at 500 °C for 1 h in a box furnace at a heating rate of 7.9 °C min$^{-1}$. The heat-treated CNT powder was then acid washed at 80 °C for 12 h in 810 g of 6 M HNO$_3$ (diluted from 60% HNO$_3$, Samchun Chemicals) under vigorous stirring. After filtration and washing with excess deionized (DI) water, the powder was treated with 720 g of 6 M HCl (diluted from 36% HCl, Samchun Chemicals) under vigorous stirring. The acid-treated CNTs were collected after drying overnight in an oven at 60 °C.

Pt$_1$/CNT catalysts with various Pt contents (Pt$_1$(X)/CNT, where X = nominal wt.% of Pt) were synthesized by solid-state mixing of CNT and Pt-macrocycle precursor followed by annealing[17]. The acid-treated

CNT (500 mg) and PtTPP (95%, Frontier Scientific) were ground in an agate mortar over 20 min until the color and texture became constant. The PtTPP contents in the precursor mixtures were 71.0, 21.6, and 2.1 mg for Pt$_1$(3)/CNT, Pt$_1$(1)/CNT, and Pt$_1$(0.15)/CNT, respectively. Subsequently, the powder mixture was pyrolyzed at 700 °C for 3 h under an N$_2$ flow (5 N, 1 L min$^{-1}$) at a heating rate of 2.1 °C min$^{-1}$. N-doped CNT was synthesized by a similar method, but 54.0 mg of TPP (1 – 3% Chlorin, Frontier Scientific), which is equivalent to 71.0 mg of PtTPP, was used as a precursor.

Two model catalysts with abundant oxygen (O-Pt$_1$(3)/CNT) or chlorine (Cl-Pt$_1$(3)/CNT) functionalities were prepared by post-treatment of Pt$_1$(3)/CNT. O-Pt$_1$(3)/CNT (150 mg) was prepared by ozone treatment at 25 °C for 1 h using an ozone generator (LAB-I, Ozonetech Inc.). Cl-Pt$_1$(3)/CNT was prepared by sequential H$_2$O$_2$ and SO$_2$Cl$_2$ treatments[50]. Pt$_1$(3)/CNT (75 mg) was mixed with a 12.7 wt.% H$_2$O$_2$ solution (1.5 L; diluted from 29–32% H$_2$O$_2$, Alfa Aesar), and the mixture was stirred at 70 °C for 2 h. The catalyst powder was collected by filtration and washed several times with DI water. Subsequently, H$_2$O$_2$-treated Pt$_1$(3)/CNT (70 mg) was dispersed in 4.2 mL of acetonitrile (99.8%, Sigma-Aldrich), and 280 mg of SO$_2$Cl$_2$ (97%, Sigma-Aldrich) was added. The mixture was stirred for 2 h at 75 °C and subsequently, heated and refluxed for 5 h at 75 °C. Cl-Pt$_1$(3)/CNT was collected via filtration and washed several times with DI water.

### Physical characterizations

HAADF-STEM images were obtained using a Titan$^3$ G2 60-300 microscope (FEI Company) equipped with a double-sided spherical aberration (Cs) corrector operated at an accelerating voltage of 200 kV. XRD patterns were obtained using a high-power X-ray diffractometer (D/MAX2500V/PC, Rigaku) equipped with Cu Kα radiation operated at 40 kV and 200 mA. The XRD patterns were measured in the 2θ range from 10° to 90° at a scan rate of 2° min$^{-1}$. XRD samples were prepared by pelletizing 50 mg of the catalyst in a sample holder (13 mm in width) under 8 tons of hydraulic pressure. XPS measurements were performed using a K-Alpha spectrometer (Thermo Fisher Scientific) equipped with a monochromatic Al Kα X-ray source (1486.6 eV). XPS Pt 4$f$ and N 1$s$ spectra were analyzed using the XPSPeak41 software with a mixed Gaussian (70)–Lorentzian (30) function after applying Shirley-type background correction. The spin-orbit components of the XPS Pt 4$f$ spectra were fixed at 3.34 eV. To quantify the Pt content in the catalysts, a microwave digestion system (Mars 6, CEM) was used to completely dissolve Pt in aqua regia (36% HCl:60% HNO$_3$ = 3:1, v/v) at 220 °C for 40 min (600 W, heating rate of 6.7 °C min$^{-1}$). Subsequently, the resulting solution was analyzed by ICP-OES (700-ES, Varian).

Ex situ Pt L$_3$-edge XAS spectra were collected at the 6D beamline of the Pohang Accelerator Laboratory (PAL). The XAS spectra of the samples were obtained in the transmission mode after pelletizing the catalysts in a sample holder (1 cm in width). Background removal and normalization of the absorption coefficient for XANES spectra and fitting for the Fourier-transformed $k^3$-weighted EXAFS spectra were performed using the Athena and Artemis software with 1.1–1.2 of Rbkg in a Hanning-type window[61]. A modified Victoreen equation was applied to normalize the post-edge signal to the step of one[62]. For the quantitative XANES fitting, a combination of the Gaussian function and arctangent function was used (Eq. (1)–(3))[63].

$$I(E) = a_1(E) + g_1(E) + g_2(E) \qquad (1)$$

$$a_{i=1}(E) = h_i[0.5 + \pi^{-1} \tan^{-1}\{(E - E_i)/w_i\}] \qquad (2)$$

$$g_{j=1,2}(E) = h_j \exp[-\ln2(E - E_j)^2\}/w_j^2] \qquad (3)$$

where $I$, $E$, $a(E)$, $g(E)$, $h$, $w$, $E_i$, and $E_j$ represent the normalized intensity, X-ray energy (eV), arctangent function, Gaussian function, the height

of peak, the width of peak, inflection point (eV), and peak position (eV), respectively. The $a_1(E)$ indicated a fundamental transition from 2$p$ to 5$d$ orbitals. The $h_1$ in $a_1(E)$ should be fixed as 1, as all fittings were conducted with normalized XANES spectra. Two Gaussian functions ($g_1(E)$ and $g_2(E)$) were used to fit the $p \to d$ transitions, indicating the WL peak and the post-edge peak, respectively (Supplementary Fig. 4). The post-edge peaks generally exhibit the electron transfer from 2$p$ to the unoccupied $d$-orbitals hybridized with ligands[64]. The average oxidation numbers were estimated using the equation ($y = 0.5367 x + 0.6549$, $x$ is the normalized WL area) by interpolating the plot of Pt references in our previous report[30]. For EXAFS fitting, the amplitude reduction factor ($S_0^2$) of Pt was fixed at 0.84 after calibration using a standard Pt foil, while the details of fitting were listed (Supplementary Table 2). Crystallographic data for the PtTPP molecule were used for multishell fitting with the first-shell of Pt−N and the second-shell of Pt···C[65].

### Electrochemical characterizations

Electrochemical measurements were conducted in a conventional three-electrode H-type cell using a potentiostat (VMP3, Bio-Logic Science Inc.). A homemade rotating disk electrode (RDE) with mirror-polished glassy carbon (5 mm diameter), Pt wire (CE-100, EC Frontier), and saturated Ag/AgCl (RE-1TA, EC Frontier) electrodes were used as the working, counter, and reference electrodes, respectively. The counter electrode was separated from the working and reference electrodes using a glass frit. To prevent unexpected contamination from the reference electrode[66], it was doubly separated from the electrolyte using a glass tube equipped with a glass frit. Ar-saturated 0.1 M HClO$_4$ solutions with and without 1 M NaCl (or 1 M NH$_4$Cl), which were prepared using DI water (≥18.2 Ω, Arium Mini, Sartorius), concentrated HClO$_4$ solution (70%, Sigma-Aldrich), NaCl (99%, Sigma-Aldrich), and NH$_4$Cl (99.5%, Sigma-Aldrich), were used as electrolytes. All potentials are given relative to the RHE scale after calibration of the reference electrode with a Pt wire electrode in an H$_2$-saturated electrolyte before each electrochemical measurement.

A thin-film electrode was fabricated by drop-casting the catalyst ink (10 µL) onto an RDE. The catalyst loading was 100 µg cm$^{-2}$. The catalyst ink was prepared by dispersing 5 mg of the catalyst in a mixed solution of DI water (2122 µL), isopropyl alcohol (374 µL), and 5 wt.% Nafion solution (50 µL). Before measuring the CER activity, the working electrode was electro-chemically activated by 50 cycles of CV in the potential range of 0.05–1.2 V$_{RHE}$ at a scan rate of 500 mV s$^{-1}$ in an Ar-saturated 0.1 M HClO$_4$ electrolyte. CER polarization curves were obtained in the potential range of 1.0–1.6 V$_{RHE}$ at a scan rate of 10 mV s$^{-1}$ in Ar-saturated 0.1 M HClO$_4$ with 1 M NaCl (or 1 M NH$_4$Cl). During the measurements, the working electrode was rotated at 1600 rpm using a rotor (RRDE-3A, ALS). Durability tests were performed using 500 CV cycles in the potential range of 1.0–1.6 V$_{RHE}$ at a scan rate of 100 mV s$^{-1}$. In this study, the onset potential of the CER was defined as the potential at 1 mA cm$^{-2}$ during CER polarization. All electrochemical results were shown after 85% $iR$ compensation correction, which was conducted by electro-chemical impedance spectroscopy (EIS) at a fixed potential of 0.9 V$_{RHE}$ in the frequency range of 100 kHz–1 Hz with a potential amplitude of 10 mV.

The electrochemical Cl$_2$ formation was analyzed by chron-oamperometry (CA) using a RRDE (012613, ALS). The catalyst loading on the disk electrode was 100 µg cm$^{-2}$. The CER selectivity was mea-sured for 120 s at an electrode rotation speed of 1600 rpm; this step was repeated five times with an intermittent break of 1 min. The applied disk potential was adjusted to generate a current density of ≥10 mA cm$^{-2}$, but the applied Pt ring potential was fixed at 0.95 V$_{RHE}$[67]. Prior to the RRDE study, the background currents of the disk and ring electrodes were stabilized at 0.95 V$_{RHE}$ with an electrode rotation of 1600 rpm. The net CER current ($i_{CER}$) at the disk electrode and CER

selectivity were calculated using the following equations.

$$i_{CER} = |\frac{i_r}{N}| \qquad (4)$$

$$Cl_2 \text{ selectivity}(\%) = 100 \cdot \frac{2 \cdot i_{CER}}{i_d + i_{CER}} = 100 \cdot \frac{2 \cdot |\frac{i_r}{N}|}{i_d + |\frac{i_r}{N}|} \qquad (5)$$

where $i_r$, $N$, and $i_d$ denote the background-corrected ring current, collection efficiency (0.35–0.37, calibrated using $K_3[Fe(CN)_6]$), and background-corrected disk current, respectively.

### Online EFC/ICP-MS measurements

Online Pt dissolution was analyzed by ICP-MS (iCAP RQ, Thermo-Fisher Science) coupled with a homemade EFC. The EFC was composed of a U-shaped channel (1 mm diameter) and two openings (3 mm diameter). On one opening side, a mirror-polished 3 mm glassy carbon electrode (002012, ALS) made electrochemical contact with the electrolyte (Supplementary Fig. 9). On the other opening side, a 3 mm Teflon tube, which was sealed with a polytetrafluoroethylene (PTFE) membrane (WP-020-80, Sumitomo Electric Ind., Ltd.) at one end, was approached to the working electrode to extract any evolved gas products by vacuum. The counter electrode was a graphite rod separated from the electrolyte by a Nafion 115 membrane (DuPont). The reference electrode was a saturated Ag/AgCl electrode that was directly connected to the outlet of the EFC. The electrolyte was Ar-saturated 0.1 M HClO₄ with 1 M NH₄Cl, which continuously flowed to the EFC at a flow rate of 400 µL min⁻¹ (Note: we avoided using 1 M NaCl due to significant damage on the sampler and skimmer cones of the ICP-MS instrument; Supplementary Fig. 10). Prior to introducing the electrolyte to the ICP-MS instrument, it was mixed with 0.5 M HNO₃ containing 5 ppb ¹⁸⁷Re as an internal standard at a mixing ratio of 1:1 using a Y-connector. Online Pt dissolution was estimated using the ratio of ¹⁹⁵Pt to ¹⁸⁷Re signals during the electrochemical treatments. The catalyst loading on the working electrode was 100 µg cm⁻². After stabilizing the online ICP-MS signals for 30 min at an open-circuit potential (OCP), the working electrode was electrochemically activated by 50 CV cycles at a scan rate of 500 mV s⁻¹ in the potential range of 0.05–1.2 $V_{RHE}$. Subsequently, Pt dissolution was monitored over 500 CV cycles at a scan rate of 100 mV s⁻¹ in the potential range of 1.0–1.6 $V_{RHE}$.

### Online EFC/DEMS measurements

The gaseous products were analyzed online by DEMS. The EFC connected to a mass spectrometer (Max 300 LG, Extrel) was constructed for DEMS analysis. The EFC equipped a U-shaped channel with a 10 mm opening diameter at the bottom, which allowed for electrical contact with the 3 mm glassy carbon working electrode (A-011169, Bio-Logic). To collect volatile and gaseous products, a porous PTFE membrane (WP-010-80, Sumitomo Electric Ind., Ltd.) was positioned approximately 100 µm above the working electrode. The graphite tube (inner diameter = 3 mm) and saturated Ag/AgCl reference electrodes were electrically connected to the outlet of the EFC. The catalyst loading on the working electrode was 560 µg cm⁻². Ar-saturated 0.1 M HClO₄ and 0.1 M HClO₄ + 1 M NaCl were used as electrolytes. The online EFC/DEMS measurements were conducted using two consecutive CVs at a scan rate of 5 mV s⁻¹ in the potential range of 1.0–1.6 $V_{RHE}$. The electrolyte flow rate was set to 0.07 mL min⁻¹ using a syringe pump (TYD01-01, LEADFLUID). The mass signals of $O_2^+$ ($m/z = 32$), $Cl^+$ ($m/z = 35$), $HCl^+$ ($m/z = 36$), and $OCl^+$ ($m/z = 51$) were collected simultaneously during the electrode polarizations.

### In situ XAS measurements

The in situ Pt $L_3$-edge XAS measurements were conducted at the 8 C beamline of the PAL, utilizing a flow-type in situ XAS cell equipped with an electrolyte flow channel and a window for X-ray radiation. The window was a carbon-coated Kapton film (200RS100, DuPont) with a thickness of 0.05 mm and an area of 0.503 cm², which was used as a working electrode. A thin-film electrode was fabricated by drop-casting of the concentrated catalyst ink with targeted loadings of 5 mg cm⁻² for Pt₁(3)/CNT and Pt₁(1)/CNT and 7 mg cm⁻² for Pt₁(0.15)/CNT. Pt wire counter and Ag/AgCl reference electrodes were directly connected to the outlet of the electrolyte stream. The fluorescence mode was used to collect the XAS spectra after calibration with a Pt foil reference. Ar-saturated 0.1 M HClO₄ + 1 M NaCl electrolyte was used as electrolyte, and the spectra were collected at the OCP and 1.45 $V_{RHE}$, respectively. Background removal and normalization of the absorption coefficient for XANES spectra and fitting for the Fourier-transformed $k^3$-weighted EXAFS spectra were performed using the Athena and Artemis software with 1.2 of Rbkg in a Hanning-type window. For EXAFS fitting, the $S_0^2$ value of Pt was fixed at 0.84 after calibration using a standard Pt foil, while the details of fitting were listed (Supplementary Table 3).

### DFT calculations

Electronic structure calculations for periodically replicated appropriate models were performed using the Vienna ab initio simulation package (VASP 5.4.1) based on the framework of DFT[68]. Full computational details can be found in Supplementary Note 3.

## Data availability

The data generated in this study have been deposited in the Zenodo repository database without accession code [https://zenodo.org/record/7936631#.ZGH5O3ZByUk][69].

## Code availability

The DFT codes generated in this study have been deposited in the Zenodo repository database without accession code [https://zenodo.org/record/7936174#.ZGHd2XZByUk][70].

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

## Acknowledgements

This work was supported by the National Research Foundation (NRF) of Korea grant funded by the Ministry of Science and ICT (MSIT) (Nos. 2021R1A5A1030054 and 2022K1A4A7A04095893 to C.H.C.; 2019M3D1A1079306 and 2021R1A2C2007495 to S.H.J.). K.S.E. is associated with the CRC/TRR247: "Heterogeneous Oxidation Catalysis in the Liquid Phase" (Project number 388390466-TRR 247), the RESOLV Cluster of Excellence, funded by the Deutsche Forschungsgemeinschaft under Germany's Excellence Strategy – EXC 2033–390677874–RESOLV, and the Center for Nanointegration (CENIDE). Experiments at PLS-II were supported in part by MSIT and POSTECH.

## Author contributions

C.H.C., S.H.J. and K.S.E. conceived and directed the project. T.L. synthesized and characterized the catalysts. J.C. and H.K. performed the electrochemical studies. L.M., F.V. and F.I. conducted computational calculations. J.K., S.L., J.H.L., G.Y.J. and K.S.L. contributed to part of the experimental and theoretical studies. J.C., T.L., H.K., K.S.E., S.H.J. and C.H.C. wrote the manuscript with contribution from all authors.

## Competing interests

The authors declare no competing interests.
