## [Peer review file · Nature Communications]

REVIEWER COMMENTS

Reviewer #1 (Remarks to the Author):

In this paper, the authors have synthesized a carbon nanotube supported low-coordinated PtII single-atom catalyst with different Pt contents and coordination structures for chlorine evolution reaction (CER). The as-synthesized Pt1(3)/CNT catalyst exhibits outstanding CER performance with high selectivity compared with competed OER process, then a series of experiments have confirmed that the tailored coordination geometry of Pt sites is responsible for the high CER activity of the catalysts by changing the content of Pt elements. Overall, the logic and organization of this article is good and the characterization is also comprehensive. I agree that this paper fill well with the scope of the journal. Therefore, I recommend it should be published after some questions solved.

1. The CER activity of Pt1(3)/CNT catalyst lacks comparison with other literature, which should be summarized and provided for better comparison.
2. The background of CER is missing. The authors are suggested to enhance the introduction on the background as well as the current progress of CER, for example: DOI: 10.1002/anie.202200366; etc.
3. In Supplementary Figure 4, what does the black line and the hollow circle represent should be marked, and the calculation formula of O_x should be given.
4. The author said that the Pt-N3(V) configuration is lack of stability. I think that the durability test of Pt1(1)/CNT catalyst should be also provided for further explanation. Besides, based on the conclusion, how to increase the stability of Pt1/CNT catalyst?
5. The authors are suggested to enhance the discussion on the structure-performance relationship of active sites at atomic scale. Some references might be helpful for your discussion: DOI: 10.1007/s12274-022-4371-x; etc.
6. What is the activity and coordination structure of Pt1/CNT catalyst if the Pt dosage is further reduced?

Reviewer #2 (Remarks to the Author):

The article entitled "Importance of Broken Geometric Symmetry of Single-Atom Pt Sites for Efficient Electrocatalysis" by Cho reported an interesting case for electrocatalysts design through engineering the symmetry of the coordination environment of Pt active site. This design is innovative. The authors used the single-atom Pt catalysts for chlorine evolution reaction as the model system. The chlorine evolution reaction is important due to the wide application of chlorine industry gas. The current bench catalyst-

based chlorine production is energy intensive. The catalyst design is crucial to address this issue. In this paper, the authors change the loading of Pt to synthesize the Pt SACs with different coordination environments of the Pt active site. Combining the experimental characterisation results with the DFT computations, the authors found the coordination environment of Pt not only changes the reactivity but also alters the reaction mechanisms and stability. Their results reveal that the Pt-N3V may be the origin of the high performance of these CER catalysts.

The results are very interesting. I think this paper can be accepted after addressing the following issues:

- 1) In Supplementary Fig. 12, the Pourbaix diagram between U and pH was drawn to demonstrate the axially unoccupied Pt site is favoured through the comparison with *OCl . After that, the Pourbaix-like diagrams between ΔG and overpotential (I guess) indicate that O^* was energetically preferred on Pt-N3 and Pt-N3V. Why were two different kinds of diagrams used here for the comparison? Why not study O^* on Pt-N4 to check whether O^* is also favoured on Pt-N4?
- 2) It seems that the authors used a cluster by using hydrogen to saturate the dangling bonds to simulate graphene. If so, it is not necessary to use a very dense (4x4x1) k-point mesh. Gamma-only k-point should be okay. By using the cluster model, the size of the cluster may affect the energies. Did the authors use a large cluster to demonstrate that the reported values are already converged? This is important because the energy difference shown in Fig. 4 is very small.
- 3) The authors used an 'optimal' kinetic energy cutoff of 415 eV. This choice needs to be further clarified. For this kind of computation including O and N species, it often needs a cutoff energy higher than 500 eV.
- 4) Equation S25 needs to be rewritten because the ΔG_{stab} is energy. However, U_{diss} is potential. Based on my understanding, the higher U_{diss} suggests higher stability. If it is true, the U_{diss} values suggest that Pt-N4 is more energetically stable. So, could the authors explain why Pt-N3V is first formed when the Pt loading is low?
- 5) The authors highlighted the d8 metal ions. This was supported by some recent theoretical studies, which demonstrated that the Ni(II) can act as the active sites for chlorine evolution reactions (<https://doi.org/10.1021/acs.jpcc.2c01593>; <https://doi.org/10.1016/j.jelechem.2022.116071>). My question is whether the same strategy by manipulating the coordination environment of active sites can also be used for other metal cations without d8 electronic configurations. If it is only for metal cations with d8 electronic configurations, could the authors provide a brief explanation?

Reviewer #3 (Remarks to the Author):

Comments to authors:

In this work, the authors synthesized low-coordinated PtII species in carbon-based Pt SACs for CER and OER, aimed at revealing the roles of broken geometric symmetry of Pt SACs by combining operando ICP-MS and ex-situ EXAFS. However, Pt element is not the most active and important for CER and OER but for HER or ORR, which greatly reduces the importance of this study. Combining with some other critical defects of this manuscript, we are sorry that this work cannot be published in Nature Communications. Detailed comments are given below:

1. The authors should demonstrate the importance of the usage of Pt in CER and OER because Pt is not that important for these two reactions. Besides, possible lab-scale applications containing CER should be conducted.
2. The products from CER and OER should be further identified by mass spectrum or chromatography.
3. The stability of the optimum catalyst is poor, only 6000 s (less than 2 h), hindering its possible application.
4. The EXAFS fitting results of aged Pt1(3)/CNT in Table S1 is not good, where the R factor (6%) is much larger than the reasonable requirement (< 2.5%).
5. Besides operando ICP-MS, operando Pt-L3 XANES and EXAFS as well as operando XPS of O 1s and Pt 4f should be performed to precisely identify catalyst operando behaviors.
6. Necessary experiments and calculations should be conducted to support the claim of “This study suggests general guidelines for achieving high electrocatalytic performance of carbon-based SACs based on all other d8 metal ions, e.g., NiII and PdII as well as PtII”.

Reviewer: 1

In this paper, the authors have synthesized a carbon nanotube supported low-coordinated PtII single-atom catalyst with different Pt contents and coordination structures for chlorine evolution reaction (CER). The as-synthesized Pt1(3)/CNT catalyst exhibits outstanding CER performance with high selectivity compared with competed OER process, then a series of experiments have confirmed that the tailored coordination geometry of Pt sites is responsible for the high CER activity of the catalysts by changing the content of Pt elements. Overall, the logic and organization of this article is good and the characterization is also comprehensive. I agree that this paper fill well with the scope of the journal. Therefore, I recommend it should be published after some questions solved.

Response: We are grateful for the reviewer's positive and constructive comments on the present work and the insightful questions raised. Detailed responses to the reviewer's comments are noted below.

1. The CER activity of Pt1(3)/CNT catalyst lacks comparison with other literature, which should be summarized and provided for better comparison.

Action: In the revised manuscript, we additionally compared the CER activity with other literature (copied below) and provided a comparison of CER activity (Table R1) in the revised Supplementary Information.

[Main manuscript/RESULTS] *Notably, this CER activity outperforms that of the commercial DSA (Supplementary Fig. 7) and most of the previously reported CER catalysts (Supplementary Table 1)*³⁵⁻⁴⁰.

Table R1: Comparison of CER activities and operational conditions for previously reported CER catalysts in acidic media.

Catalysts	η @ 10 mA cm ⁻² (mV)	CER operation conditions	Precious metal contents	Ref.
Pt ₁ (3)/CNT	50	0.1 M HClO ₄ + 1 M NaCl (25 °C)	3wt.% Pt	This work
Commercial DSA, Ru-Ti-Ir/Ti (Siontech, Korea)	105	0.1 M HClO ₄ + 1 M NaCl (25 °C)	NA	This work
RuO ₂ (110)	140	0.1 M HClO ₄ + 1 M NaCl (25 °C)	76wt.% Ru	(1)
Commercial DSA, Ru-Ti-Ir/Ti (Covestro, Germany)	90	3.0 M NaNO ₃ + 1.0 M NaCl (pH 3 adjusted by adding HCl, 25 °C)	NA	(2)
Ru _{0.3} Sn _{0.7} O ₂ /Ti	32	3.5 M NaCl (pH 3, 80 °C)	21wt.% Ru	(3)
Mesoporous Ru-Ir/TiO ₂	140	4.0 M NaCl (pH 3, 40 °C)	7.5wt.% Ru 7.5wt.% Ir	(4)
Ir ₁ -TiC	30	Cl ₂ -saturated 4 M NaCl (pH 2 adjusted by adding HCl, 25 °C)	0.41wt.% Ir	(5)
RuTiO _x	130	Cl ₂ -saturated 4 M NaCl (pH 2 adjusted by adding HCl, 25 °C)	NA	(6)

(1) *ACS Catal.* **7**, 2403 (2017), (2) *Phys. Chem. Chem. Phys.* **16**, 13741 (2014), (3) *Phys. Chem. Chem. Phys.* **14**, 7392 (2012), (4) *ACS Catal.* **3**, 1324 (2013), (5) *Angew. Chem. Int. Ed.* **61**, e202200366 (2022), and (6) *Energy Environ. Sci.* **12**, 1241 (2019)

2. The background of CER is missing. The authors are suggested to enhance the introduction on the background as well as the current progress of CER, for example: DOI: 10.1002/anie.202200366; etc.

Action: In the revised manuscript, the background and current progress of CER have been additionally introduced. Also, the suggested paper was introduced in the revised manuscript as a reference #39.

[Main manuscript/INTRODUCTION] *The chlorine evolution reaction (CER) is exemplified as a model reaction, of which the product, Cl₂, is practically important due to extensive applications in the chemical industry, typically produced via the chlor-alkali process with the Ru/Ir-based dimensionally stable anode (DSA) electrodes*^{28,29}.

3. In Supplementary Figure 4, what does the black line and the hollow circle represent should be marked, and the calculation formula of Ox should be given.

Action: Hollow circles and solid black lines indicated the raw and fitted XANES spectra, respectively. For clear representation, we specified the corresponding legends in the revised Supplementary Fig. 4 (Fig. R1).

For the details of quantitative XANES fitting, we utilized the ‘peak fitting’ function of the Athena application in the Demeter software. The values of oxidation numbers were obtained by interpolating the normalized area of the white line (WL) peak (red filled area of 11,560–11,572 eV; Fig. R1) into the equation ($y = 0.5367x + 0.6549$; x is area of the normalized WL) calculated with Pt references [*ACS Catal.* **11**, 12232 (2021)] (Fig. R2). In detail, the WL area in the XANES spectra can be an informative index for the d -vacancy of absorbent, *i.e.*, Pt in this case, by considering that the Pt L₃-edge XANES spectra originate from the electronic transition from $2p$ to $5d$ orbitals. A baseline of p -, d transition was established by an arctangent function (Edge jump), while the WL was fitted with a Gaussian function (g_1 , WL peak). A shoulder peak (Post-edge) beyond the WL around 11,573 eV indicated the electron transfer from $2p$ to the unoccupied $5d$ -orbitals hybridized with ligands [*Chem. Phys. Lett.*, **316**, 495 (2000)]. Thus, this is reasonable to fit the area with another Gaussian function (g_2 , Post-edge peak). We also added the details of the quantitative XANES fitting in the Experimental details section.

Fig. R1: a–c Pt L₃-edge XANES spectra of Pt₁(3)/CNT (a), Pt₁(1)/CNT (b), and Pt₁(0.15)/CNT (c). The results show that the average oxidation state of the catalysts is Pt^{II}. The XANES WL fitting was conducted using the interpolation equation of Pt references in our previous report [*ACS Catal.* **11**, 12232 (2021)]. The fitting consists of one arctangent function ($a_1(E)$) and two Gaussian functions ($g_1(E)$ and $g_2(E)$) by the following equation: $I(E) = a_1(E) + g_1(E) + g_2(E)$.

Fig. R2: Average oxidation numbers of Pt₁(3)/CNT, PtTPP precursor, K₂PtCl₄, PtO₂, and H₂PtCl₆ plotted by the results from normalized WL area, which was obtained by the quantitative XANES fitting. The equation of $y = 0.5367x + 0.6549$ was used to calculate Pt oxidation number.

4. (4a) The author said that the Pt-N₃(V) configuration is lack of stability. I think that the durability test of Pt₁(1)/CNT catalyst should be also provided for further explanation. (4b) Besides, based on the conclusion, how to increase the stability of Pt₁/CNT catalyst?

Note: As one comment and one question were raised in the reviewer's comment 4, we would like to separate them into (4a) and (4b) to respond more clearly.

Response to comment (4a): In addition to the previous durability tests with Pt₁(3)/CNT and Pt₁(0.15)/CNT, we additionally evaluated the durability of Pt₁(1)/CNT. After the iterative 500 CVs from 1.0 to 1.6 V_{RHE}, the CER activity of Pt₁(1)/CNT decreases by 65% from 49 to 17 mA cm⁻² at 1.45 V_{RHE} (Fig. R3a). Online EFC/ICP-MS study revealed that total Pt loss after the 500 CVs is 41%, which also hardly corresponds to the CER activity decay (Fig. R3b), as Pt₁(3)/CNT. Taking all the results into account (Fig. R3c), Pt₁(1)/CNT, which has a moderate proportion of the Pt-N₃(V) site (19%), exhibits intermediate CER activity loss and Pt dissolution amount compared to those of Pt₁(3)/CNT (7% Pt-N₃(V) proportion) and Pt₁(0.15)/CNT (~100% Pt-N₃(V) proportion). This successfully strengthens our conclusion that Pt-N₃(V) is a main catalytic site but more labile than Pt-N₄.

Fig. R3: a,b Comparison of the CER activity decrement of Pt₁(1)/CNT (**a**) and Pt loss measured by online EFC/ICP-MS (**b**) during durability test. **c** A summary of CER activity and Pt losses of Pt₁(X)/CNTs (X = 0.15, 1, and 3) after durability test.

Action to comment (4a): The stability of Pt-N₃(V) sites was further discussed by comparing CER activity and Pt losses of Pt₁(X)/CNTs (copied below), and durability test results of Pt₁(1)/CNT were newly introduced in the revised Supplementary Information (Supplementary Fig. 16).

[Main manuscript/RESULTS] *In addition, the Pt loss of Pt₁(X)/CNTs (X = 0.15, 1, and 3) after 500 CVs*

becomes intensified as the proportion of the Pt–N₃(V) increases (Supplementary Fig. 16).

Response to question (4b): After concluding that labile Pt–N₃(V) is a main catalytic site, our interest was also oriented towards answering the same question — how to improve its operational longevity. Herein, we would like to briefly introduce two strategies for mitigating CER deactivation that we are currently working on. This information is intended only for review.

The first strategy involves shifting the equilibrium of the Pt dissolution reaction by adding a small amount of Pt ions into the electrolyte. By applying Le Chatelier’s principle to the Pt dissolution reaction “[Pt] ↔ Pt²⁺ + ze⁻ + []”, where [] denotes the empty pocket of Pt–N₄ and Pt–N₃(V) moieties”, increasing chemical activity of dissolved Pt ion in the electrolyte can tune the equilibrium between [Pt] and Pt²⁺, possibly lowering Pt dissolution rate. Based on this conjecture, we performed a durability test for the Pt₁(3)/CNT in the presence of Pt(IV) ion in the electrolyte during iterative CER CV cycles from 1.0 to 1.6 V_{RHE}. In the presence of additional Pt(IV), the CER activity decay was considerably mitigated during the durability test (Fig. R4a), compared to the absence of Pt ion in the electrolyte (Fig. R4b). The possible artifact resulting from Pt ion deposition into the metallic Pt on the catalysts, which could serve as an alternative active site for CER, can be ruled out because the lower potential limit of CV is limited to 1.0 V_{RHE}, at which electroplating Pt(IV) to Pt(0) hardly occurs [*Electrochim. Acta* **176**, 65 (2015)]. The CER activity at 1.45 V_{RHE} decreases by 40% from 51 to 30 mA cm⁻² in the presence of Pt(IV), suggesting that controlling the equilibrium of the Pt dissolution reaction could be one of the possible strategies for enhancing the durability of the Pt SACs (Fig. R4c). However, this strategy will not be practically feasible as it requires a continuous supply of expensive precious noble metal ions into the feed stream (electrolyte and also brine).

Fig. R4: a,b The durability tests of Pt₁(3)/CNT performed by measuring CER polarization curves during iterative 500 CVs from 1.0 and 1.6 V_{RHE} in the presence (a) and absence of Pt(IV) ion (b) in the electrolytes. c Comparison of the CER activity decrement in the absence and presence of Pt(IV) ion.

The content here is redacted.

Action to question (4b): We newly emphasized the importance of improving the stability of the labile Pt–N₃(V) sites to achieve operational longevity.

[Main manuscript/RESULTS]...*Otherwise, more practical approaches, such as engineering the catalyst-electrolyte interface or finding optimal operating conditions (e.g., well-regulated potential excursions) to promote the longevity of labile Pt–N₃(V) sites, will be the subject of another future research direction.*

5. The authors are suggested to enhance the discussion on the structure-performance relationship of active sites at atomic scale. Some references might be helpful for your discussion: DOI: 10.1007/s12274-022-4371-x; etc.

Action: To deepen our discussion on the structure-performance relationship, the following sentence was newly added in the revised manuscript (copied below). The suggested paper and other interesting previous works were also introduced in the revised manuscript as references #12 and #20–23.

[Main manuscript/INTRODUCTION] *However, in theory, these model structures predict substantially weakened axial coordination and, consequently, poor catalytic activity, while broken or unsaturated ones are expected to offer more optimized axial coordination leading to high catalytic activity^{12,20-23}.*

6. What is the activity and coordination structure of Pt1/CNT catalyst if the Pt dosage is further reduced?

Response: We would like to note that, in practical experiments, it is difficult to prepare a catalyst with a nominal Pt loading lower than 0.15 wt.%. As one can find in Experimental details section, the Pt₁(0.15)/CNT catalyst was initially intended to load 0.1 wt.% Pt (2.1 mg of Pt^{II}TPP with 500 mg CNT; the molar masses of Pt and Pt^{II}TPP are 195.084 and 807.803 g mol⁻¹, respectively). The increased Pt content may be due to the loss of CNT during the physical mixing of Pt^{II}TPP and CNT. Accordingly, we supposed that loading of Pt less than 0.15 wt.% would result in considerable deviation and irreproducibility from the nominal content.

We also note that the Pt₁(0.15)/CNT catalyst was composed almost entirely of more active Pt–N₃(V) species and showed a slightly decreased CER activity. This result indicates that the Pt₁(0.15)/CNT catalyst had already reached the upper limit in active site density, where almost all Pt–N₃(V) active sites could participate in the CER electrocatalysis. Hence, we expect that the further lowering of Pt loading in the catalyst would result in the decrease of catalytic activity with diminished content of Pt–N₃(V) species.

Reviewer: 2

The article entitled "Importance of Broken Geometric Symmetry of Single-Atom Pt Sites for Efficient Electrocatalysis" by Cho reported an interesting case for electrocatalysts design through engineering the symmetry of the coordination environment of Pt active site. This design is innovative. The authors used the single-atom Pt catalysts for chlorine evolution reaction as the model system. The chlorine evolution reaction is important due to the wide application of chlorine industry gas. The current bench catalyst-based chlorine production is energy intensive. The catalyst design is crucial to address this issue. In this paper, the authors change the loading of Pt to synthesize the Pt SACs with different coordination environments of the Pt active site. Combining the experimental characterisation results with the DFT computations, the authors found the coordination environment of Pt not only changes the reactivity but also alters the reaction mechanisms and stability. Their results reveal that the Pt-N3V may be the origin of the high performance of these CER catalysts.

The results are very interesting. I think this paper can be accepted after addressing the following issues:

Response: The reviewer highlighted the key insights that we wish to deliver to scientists working in the field of SACs as well as the broad readership of *Nature Communications* through this work. We greatly appreciate the positive and thoughtful comments from the reviewer. We respond point-by-point to the insightful questions raised by the reviewer.

1) In Supplementary Fig. 12, the Pourbaix diagram between U and pH was drawn to demonstrate the axially unoccupied Pt site is favoured through the comparison with *OCl . After that, the Pourbaix-like diagrams between ΔG and overpotential (I guess) indicate that O^* was energetically preferred on Pt-N3 and Pt-N3V. Why were two different kinds of diagrams used here for the comparison? Why not study O^* on Pt-N4 to check whether O^* is also favoured on Pt-N4?

Response: We thank the reviewer for pointing out this inconsistency. Indeed, we constructed a (complete) Pourbaix diagram for the Pt-N₄ model (*cf.* Supplementary Fig. 13 in the original Supplementary Information). For the two Pt-N₃(V) structures (*cf.* Supplementary Fig. 14 in the original Supplementary Information), we used a simplified representation of the Pourbaix approach by plotting ΔG as a function of applied overpotential, η .

Action: To address the criticism, we have also compiled the ΔG vs. η plot for the Pt-N₄ model (Fig. R6) in the revised Supplementary Information (Supplementary Fig. 18). This diagram indicates that the free sites, denoted as * , are energetically preferred compared to *O on Pt-N₄ up to high overpotentials. This result is consistent with the (complete) Pourbaix diagram of Supplementary Fig. 13 in the original Supplementary Information.

Fig. R6: Pourbaix-like diagrams for the **a** square planar Pt-N₄, **b** trigonal planar Pt-N₃, and **c** T-shaped Pt-N₃V models.

2) It seems that the authors used a cluster by using hydrogen to saturate the dangling bonds to simulate graphene. If so, it is not necessary to use a very dense (4x4x1) k-point mesh. Gamma-only k-point should be okay. By using the cluster model, the size of the cluster may affect the energies. Did the authors use a large cluster to demonstrate that the reported values are already converged? This is important because the energy difference shown in Fig. 4 is very small.

Response: We agree with the reviewer that the usage of Γ -point is sufficient to get converged results, also in line with having an isolated model. Indeed, test calculations for the adsorbates *, *Cl, *O, and *OCl with only Γ -point yielded variations in the total energy of at most 0.01 eV, thus indicating that there is no impact on the discussed results. Regarding the cluster model, note that the modeled Pt–N_x active center is surrounded by two carbon shells leading to a large enough model to provide converged values with respect to the cluster size. We also note that within a periodic approach, a supercell of similar size will not be large enough because the structural distortion induced by the presence of the Pt–N_x active site will be a long-range one. Using a cluster model with the C atoms at the cluster edge saturated by H avoids this problem.

Action: To support the validity of our DFT calculation models, we further elaborated on the background of experimental parameters in the revised Supplementary Information.

[Supplementary Information/DFT calculations] *...To model the Pt–N_x sites, graphene-like patch cluster models were used with two carbon rings surrounding the Pt–N_x active sites, capped with H-bonds. This avoids using an exceedingly large supercell in a periodic approach necessary to avoid the long-range distortion on graphene induced by the Pt–N_x sites while large enough to lead to converged results.*

[Supplementary Information/DFT calculations] *...the Brillouin zone was sampled using a 4×4×1 k -point Γ -centered Monkhorst-Pack grid. Using the Γ -point only results in a difference of the total energy of at most 0.01 eV.*

3) The authors used an ‘optimal’ kinetic energy cutoff of 415 eV. This choice needs to be further clarified. For this kind of computation including O and N species, it often needs a cutoff energy higher than 500 eV.

Response: We used the recommended standard PAW pseudopotentials for C, O, and N, which, for the investigated materials and solids, provide converged results for a kinetic energy cutoff of 400 eV, as suggested by the developers. The usage of 415 eV as a cutoff is justified due to the rather large unit cell. In any case, we performed test calculations for the adsorbates *, *Cl, *O, and *OCl applying 600 eV as the kinetic energy cutoff, and the total energies varied at most by 0.01 eV, thus indicating the negligible impact of a larger kinetic energy cutoff on the presented results.

Action: We additionally discussed the result of the kinetic energy cutoff of 600 eV, which is consistent with that of 415 eV, employed as an optimal kinetic energy cutoff in the present work.

[Supplementary Information/DFT calculations] *...an optimal kinetic energy cutoff of 415 eV since test calculations for the *, *Cl, *O, and *OCl adsorbates with a kinetic energy cutoff of 600 eV indicate that the total energies are affected by at most 0.01 eV.*

4) Equation S25 needs to be rewritten because the delta G_{stab} is energy. However, U_{diss} is potential. Based on my understanding, the higher U_{diss} suggests higher stability. If it is true, the U_{diss} values suggest that Pt–N₄ is more energetically stable. So, could the authors explain why Pt–N₃V is first formed when the Pt loading is low?

Response: We thank the reviewer for the discussion of Eqn. S25 in the original Supplementary Information; indeed, the elementary charge, e , was missing in this equation. ΔG_{stab} refers to free energy, and free energy and applied electrode potential, U , scale with each other, $U \sim \Delta G_{\text{stab}}$. For a reduction process, the following relationship holds — $\Delta G_{\text{stab}} = -z \cdot e \cdot U$ — whereas for an oxidation process, the correlation reads: $\Delta G_{\text{stab}} = z \cdot e \cdot U$. U_{diss} can be obtained by rearranging the latter equation, given that the corresponding demetallation process — $[\text{Pt}] + 2\text{H}_2\text{O} \rightarrow [\] + \text{PtO}_2 + 4\text{H}^+ + 4\text{e}^-$ — refers to an oxidative reaction.

We agree with the reviewer that a higher value of U_{diss} indicates higher stability under anodic reaction conditions, such as encountered with the CER. While U_{diss} may serve as a measure for the stability of the Pt–N₄ and Pt–N₃(V) models under CER conditions, it does not exactly explain the initial formation of Pt–N₃(V) for low Pt loadings. We assume that the formation of Pt–N₃(V) for low Pt loadings is governed by a kinetic reaction control in that the overall reaction barrier for the formation of Pt–N₃(V) is smaller than that for the formation of Pt–N₄. Yet, Pt–N₄ is the more stable configuration, as evident from its higher value of U_{diss} . Therefore, the closer

one gets to equilibrium, the more Pt–N₄ is formed, and the formation of Pt–N₃(V) is suppressed, resulting in a thermodynamic reaction control.

Action: We corrected the Eqn. S28 (Eqn. S25 in the original manuscript), including the elemental charge in the revised Supplementary Information as follows:

$$U_{\text{diss}} = \frac{\Delta_{\text{stab}}}{4} \quad (\text{Eqn. S28})$$

Furthermore, we additionally discussed the potential origin of the preferential formation of Pt–N₃(V) sites rather than Pt–N₄ sites at the low Pt contents, even though the former sites are less stable than the latter ones.

[Supplementary Information/DFT calculations] ...We note that the U_{diss} value cannot be used to explain why Pt–N₃(V) is first formed when the Pt loading is low. We believe that the formation of Pt–N₃(V) for low Pt loadings is governed by a kinetic reaction control, whereas with higher Pt loadings, the formation of Pt–N₄ prevails since this configuration is thermodynamically preferred (thermodynamic reaction control), as evident from its larger U_{diss} value.

5) The authors highlighted the d8 metal ions. This was supported by some recent theoretical studies, which demonstrated that the Ni(II) can act as the active sites for chlorine evolution reactions (<https://doi.org/10.1021/acs.jpcc.2c01593>; <https://doi.org/10.1016/j.jelechem.2022.116071>). My question is whether the same strategy by manipulating the coordination environment of active sites can also be used for other metal cations without d8 electronic configurations. If it is only for metal cations with d8 electronic configurations, could the authors provide a brief explanation?

Response: Thank you for the reviewer’s intriguing question on the possibility of whether other transition metal centers than d^8 configurations can have high CER performance by controlling their d -orbital configurations. Although there has been research focusing on the CER on d^8 metal centers, as the reviewer mentioned, we think that manipulating the electron configuration can also be applied to other metal species. According to previous literature [Beilstein *J. Org. Chem.* **9**, 1352 (2013)], the T-shaped Pt complex with ligand vacancy (*e.g.*, Pt–N₃(V)) can greatly stabilize its lowest unoccupied $d_{x^2-y^2}$ orbital (Fig. R7) by mixing with one p_y orbital made by the removal of an antibonding interaction. It could lower the energy barrier for the charge transfer on the Pt center with d^8 configuration for the CER, *i.e.*, the OCl pathway. We envisage that the same strategy (or a more creative idea based on organometallic chemistry) can be applied to other metals if one can finely tune the geometric structure and the corresponding electron configuration of the heterogenized metal center.

Fig. R7: Orbital diagrams of metal center with d^8 configuration in square planar (D_{4h}) complex (left) and T-shaped complex (right) with ligand vacancy. In the T-shaped complex diagram, the absence of one ligand in the y-axis stabilizes the $d_{x^2-y^2}$ orbital by mixing with the p_y orbital. The figure was taken from Beilstein *J. Org. Chem.* **9**, 1352 (2013).

Reviewer: 3

In this work, the authors synthesized low-coordinated PtII species in carbon-based Pt SACs for CER and OER, aimed at revealing the roles of broken geometric symmetry of Pt SACs by combining operando ICP-MS and ex-situ EXAFS. However, Pt element is not the most active and important for CER and OER but for HER or ORR, which greatly reduces the importance of this study. Combining with some other critical defects of this manuscript, we are sorry that this work cannot be published in Nature Communications. Detailed comments are given below:

Response: As the reviewer summarized, we have revealed non-unitary active site identities in the Pt SACs and further demonstrated that their genuine catalytic sites are Pt–N₃(V) with broken symmetry, responsible for the overall reaction kinetics. We partly agree with the reviewer’s argument because the Pt element has traditionally been perceived as not the most active and important for CER (and also OER). Obviously, the metallic Pt exhibits poorer CER performance than the commercial DSA electrodes.

However, we would like to point out that the physicochemical and electrochemical properties of metallic (or bulk) Pt and single-atom Pt ions are entirely different. It is important to distinguish one from the other for accurate discussion, and we note that the above explanation is only valid when Pt identity is metallic (or bulk Pt), not Pt SACs.

Likewise, similar examples can also be easily found in Ni and Fe (and other transition metals). Their metallic forms do not effectively catalyze the electrochemical CO₂ reduction (CO₂RR) and O₂ reduction reactions (ORR) compared to the state-of-the-art catalysts (*e.g.*, Cu (or Ag) and Pt, respectively). However, their SAC conformations have abundantly demonstrated promising electrocatalytic activities, opening a new frontier of heterogeneous catalysis and leading recent research streams [*Nat. Commun.* **8**, 944 (2017); *Science* **324**, 71 (2009)].

In the present work, our main model catalyst was Pt SAC, Pt₁(3)/CNT, on which isolated Pt ions were covalently immobilized on CNT support. This material was originally reported in the same journal in 2020 [*Nat. Commun.* **11**, 412 (2020)] and has received great attention from academic communities due to its promising CER activity superior to commercial DSA and most DSA-based electrodes (Table R2). Indeed, even in industries, we have currently collaborated with several chemical companies for potential industrial applications of our Pt SACs and discussed further technology transfer for employing them to replace current DSA electrodes to reduce production price.

Despite the outstanding performance of Pt₁(3)/CNT towards CER, from a fundamental point of view, the well-known *d*⁸ electron configuration of Pt²⁺ and its preference for being stabilized by *D*_{4h} symmetry, which has induced interesting inconsistency between experiment and theory, piqued our interest in chemical nature of genuine catalytic sites. The present work exactly aims to answer this question, and by means of well-controlled model catalysts, advanced online analyses, and theoretical calculations, we demonstrate that Pt–N₃(V) with a broken symmetry serves as a main catalytic site as previously predicted by theory.

Therefore, we, and also apparently the other two Reviewers, believe that our investigation on the Pt SACs will significantly benefit the field of SACs for their rational developments.

In response to the reviewer’s comments, we provide point-by-point responses below.

Table R2: Comparison of CER activities and operational conditions for previously reported CER catalysts.

Catalysts	η @ 10 mA cm ⁻² (mV)	CER operation conditions	Precious metal contents	Ref.
Pt ₁ (3)/CNT	50	0.1 M HClO ₄ + 1 M NaCl (25 °C)	3wt.% Pt	This work

Commercial DSA, Ru–Ti–Ir/Ti (Siontech, Korea)	105	0.1 M HClO ₄ + 1 M NaCl (25 °C)	NA	This work
RuO ₂ (110)	140	0.1 M HClO ₄ + 1 M NaCl (25 °C)	76wt.% Ru	(1)
Commercial DSA, Ru–Ti–Ir/Ti (Covestro, Germany)	90	3.0 M NaNO ₃ + 1.0 M NaCl (pH 3 adjusted by adding HCl, 25 °C)	NA	(2)
Ru _{0.3} Sn _{0.7} O ₂ /Ti	32	3.5 M NaCl (pH 3, 80 °C)	21wt.% Ru	(3)
Mesoporous Ru–Ir/TiO ₂	140	4.0 M NaCl (pH 3, 40 °C)	7.5wt.% Ru 7.5wt.% Ir	(4)
Ir ₁ -TiC	30	Cl ₂ -saturated 4 M NaCl (pH 2 adjusted by adding HCl, 25 °C)	0.41wt.% Ir	(5)
RuTiO _x	130	Cl ₂ -saturated 4 M NaCl (pH 2 adjusted by adding HCl, 25 °C)	NA	(6)
NiSb ₂ O _x	410	Cl ₂ -saturated 4 M NaCl (pH 2 adjusted by adding HCl, 25 °C)	NA	(6)

(1) *ACS Catal.* **7**, 2403 (2017), (2) *Phys. Chem. Chem. Phys.* **16**, 13741 (2014), (3) *Phys. Chem. Chem. Phys.* **14**, 7392 (2012), (4) *ACS Catal.* **3**, 1324 (2013), (5) *Angew. Chem. Int. Ed.* **61**, e202200366 (2022), and (6) *Energy Environ. Sci.* **12**, 1241 (2019)

1. The authors should demonstrate the importance of the usage of Pt in CER and OER because Pt is not that important for these two reactions. Besides, possible lab-scale applications containing CER should be conducted.

Response: Once again (as responded in the first response), the conventional belief of inferior catalytic performance of Pt elements is only valid for bulk/metallic Pt catalysts, not for Pt SACs. Our Pt SACs outperform the commercial Ru/Ir-based DSA electrode and most of the previously reported CER catalysts (Table R2). In addition, we have previously performed a preliminary lab-scale application using an H-type cell, which selectively produced Cl₂ for 12 hours in our previous work [*Nat. Commun.* **11**, 412 (2020)]. Beyond such lab-scale applications, we have currently collaborated with several chemical companies for potential industrial applications of our Pt SACs. Hence, we believe that the Pt SACs could serve as one of the viable options for electrochemical Cl₂ production in the near future.

Action: In the revised manuscript, we additionally compared the CER activity with other literature (copied below) and provided a comparison of CER activity (Table R2) in the revised Supplementary Information.

[Main manuscript/RESULTS] *Notably, this CER activity outperforms that of the commercial DSA (Supplementary Fig. 7) and most of the previously reported CER catalysts (Supplementary Table 1)*³⁵⁻⁴⁰.

2. The products from CER and OER should be further identified by mass spectrum or chromatography.

Response: In response to the reviewer's comment, we conducted the additional product analysis for the Pt₁(X)/CNTs (X = 0.15, 1, and 3) using differential electrochemical mass spectrometry (DEMS), which enables a potential-resolved quantification of gaseous or volatile products *in situ* (Fig R8). During two slow CVs, the ionic currents for *m/z* = 32 and 35 were recorded to identify the main products of OER and CER, *i.e.*, O₂ and Cl₂. In addition, we further take into account the possible Cl₂ hydrolysis before introducing into the mass spectrometer (including an electrolyte, a porous PTFE membrane, and a capillary connecting the

electrochemical flow cell and mass spectrometer) according to the following equation: $\text{Cl}_2 + \text{H}_2\text{O} \rightarrow \text{HCl} + \text{HOCl}$ [Chem. Rev. **116**, 2982 (2016); J. Electrochem. Soc. **165**, E751 (2018)]. Thus, the ionic currents for $m/z = 36$ and 51 were additionally monitored for HCl and HOCl, respectively.

For all $\text{Pt}_1(\text{X})/\text{CNTs}$, the online DEMS results reveal a predominant ionic current for $m/z = 35$ (Cl^+) in 0.1 M $\text{HClO}_4 + 1$ M NaCl electrolyte, which corresponds to the fragmentation of Cl_2 and its hydrolyzed derivatives, *i.e.*, HCl and HOCl (Fig. R8b). Concurrently, distinct ionic currents for $m/z = 36$ and 51 (HCl^+ and OCl^+ , respectively) are detected, indicating the formation of HCl and HOCl during CER (Fig. R8c,d). An insignificant ionic current for $m/z = 32$ (O_2^+) is identified, corresponding to the molecular ion of O_2 (Fig. R8a), although the $\text{Pt}_1(\text{X})/\text{CNTs}$ exhibited almost 100% selectivity toward CER based on the rotating ring-disk electrode (RRDE) measurements (Supplementary Fig. 6 in the original manuscript).

Fig. R8: a–d Online DEMS results of $m/z = 32$ (a), 35 (b), 36 (c), and 51 (d) of the $\text{Pt}_1(\text{X})/\text{CNTs}$ ($X = 0.15, 1,$ and 3) during two consecutive slow CVs in an Ar-saturated 0.1 M $\text{HClO}_4 + 1$ M NaCl electrolyte.

We attribute this marginal O_2 signal ($m/z = 32$) to results originating from the further HOCl decomposition in the vacuum system, forming O_2 as a product by the following equation: $2\text{HOCl} \rightarrow 2\text{HCl} + \text{O}_2$ [Chem. Rev. **116**, 2982 (2016); J. Electrochem. Soc. **165**, E751 (2018)]. To confirm this scenario, we performed an identical experiment in the absence of NaCl in the electrolyte and found no appreciable production of O_2 and Cl_2 during CVs (Fig. R9). Therefore, we can reasonably conclude that the O_2 signal during CER, as shown in Fig. R8a, is not a result of Faradaic reaction, *i.e.*, OER, but an artifact resulting from the Cl_2 hydrolysis and subsequent HOCl decomposition in the vacuum system of our DEMS setup.

Hence, the DEMS analysis successfully strengthens the previous RRDE results that $\text{Pt}_1(\text{X})/\text{CNTs}$ are highly selective towards CER rather than OER.

Fig. R9: a,b Online DEMS results of $m/z = 32$ (**a**) and 35 (**b**) of the $Pt_1(X)/CNTs$ ($X = 0.15, 1,$ and 3) during two consecutive slow CVs in an Ar-saturated 0.1 M HClO_4 electrolyte.

Action: We newly added a discussion on the online DEMS results in the revised manuscript (Fig. 1 and Supplementary Fig. 14). Additionally, origins of the marginal O_2 signals were further discussed in the revised Supplementary Information (Supplementary Note 2).

[Main manuscript/RESULTS] *The online differential electrochemical mass spectrometry (DEMS) measurement reveals a predominant ionic current for $m/z = 35$ (Cl^-) in $0.1\text{ M HClO}_4 + 1\text{ M NaCl}$ electrolyte (Fig. 1b), which corresponds to the fragmentation of Cl_2 and its hydrolyzed derivatives from the following equation: $Cl_2 + H_2O \rightarrow HCl + HOCl$ ^{33,34}. Concomitantly, distinct ionic currents for $m/z = 36$ and 51 (HCl^+ and OCl^+ , respectively) are detected, indicating the formation of HCl and $HOCl$ during CER. An insignificant ionic current for $m/z = 32$ (O_2^-) may attribute to the further decomposition of $HOCl$ forming O_2 via the following equation: $2HOCl \rightarrow 2HCl + O_2$ (cf. Supplementary Note 2 for a detailed discussion of non-Faradaic O_2 formation)^{33,34}. In NaCl-free 0.1 M HClO_4 electrolyte, the O_2 is not detected within the CER-relevant potential window, inferring that the $Pt_1(3)/CNT$ catalyzes CER selectively against competitive OER (Fig. 1c). The DEMS results are further corroborated by the rotating ring disk electrode (RRDE) measurement, which exhibits approximately 100% CER selectivity (Supplementary Fig. 6).*

Fig. 1: CER performance of $Pt_1(3)/CNT$. **a** CER polarization curves of the CNT, N-doped CNT, and $Pt_1(3)/CNT$ catalysts obtained in Ar-saturated 0.1 M HClO_4 with 1 M NaCl . The polarization curve of $Pt_1(3)/CNT$ measured in an NaCl-free electrolyte is also shown (dotted line). **b,c** Online DEMS results of $m/z = 32, 35, 36,$ and 51 of

$Pt_1(3)/CNT$ during two consecutive slow CVs obtained in Ar-saturated 0.1 M $HClO_4$ with 1 M NaCl (b) and NaCl-free electrolytes (c). d The polarization curves of $Pt_1(3)/CNT$ in Ar/ CO -saturated 0.1 M $HClO_4$ with and without 1 M NaCl. e The durability test of $Pt_1(3)/CNT$ performed by measuring CER polarization curves during iterative 500 CVs from 1.0 and 1.6 V_{RHE} . f Comparison between the CER activity decrement and Pt loss measured during the durability test.

Supplementary Fig. 14: a–d Online DEMS results of $m/z = 32, 35, 36,$ and 51 of $Pt_1(1)/CNT$ (a,b) and $Pt_1(0.15)/CNT$ (c,d) during two consecutive slow CVs obtained in Ar-saturated 0.1 M $HClO_4$ with 1 M NaCl (a,c) and in NaCl-free electrolytes (b,d).

3. The stability of the optimum catalyst is poor, only 6000 s (less than 2 h), hindering its possible application.

Response: In the present work, $Pt_1(X)/CNT$ electrodes experienced harsh potential excursions from 1.0 to 1.6 V_{RHE} to accelerate their degradation rates [J. Electrochem. Soc. **165**, F3148 (2018)]. Under realistic operating conditions, e.g., the constant voltage, the $Pt_1(3)/CNT$ exhibits promising durability with 72% of initial activity over 12 h (under constant potential at an initial current density of 10 mA cm^{-2}), which is comparable with that of a commercialized DSA electrode (Fig. R10).

Fig. R10: Durability tests for the $Pt_1(3)/CNT$ and commercialized DSA electrode under constant potential at an initial current density of 10 mA cm^{-2} .

Action: In the revised manuscript, we additionally described the rationale for adopting iterative CVs as a means

to accelerate catalyst degradation and, in the revised Supplementary Information, further provided the CER stability of Pt₁(3)/CNT under constant potential conditions relevant to real CER electrolysis, which presents promising CER stability.

[Main manuscript/RESULTS] We adopted iterative CVs for accelerating catalyst degradation, as this catalyst exhibits promising stability under constant potential conditions relevant to real CER electrolysis conditions (Supplementary Fig. 8).

4. The EXAFS fitting results of aged Pt₁(3)/CNT in Table S1 is not good, where the R factor (6%) is much larger than the reasonable requirement (< 2.5%).

Response: In the EXAFS fitting of the aged Pt₁(3)/CNT sample, we found that the Pt···C shell at 2.5 Å was missed, which could generate a high R-factor. We conducted a new EXAFS fitting, including the Pt···C shell (Fig. R11 and Table R3). As a result, the coordination numbers of Pt–N (4.0) and Pt–Cl (0.6) remain the same as the previous results, of which the R-factor for three-shell fitting is only 1.3%.

Fig. R11: The k^3 -weighted Pt L₃-edge EXAFS spectrum and fitted curves of Pt₁(3)/CNT after 500 CVs in the potential range of 1.0–1.6 V_{RHE} in Ar-saturated 0.1 M HClO₄ with 1 M NaCl. The detailed fitting parameters are provided in Table R3.

Table R3: Summary of EXAFS fitting parameters of aged Pt₁(3)/CNT.

Sample	k range	R range	Shell	CN	R (Å)	σ^2 (10^{-3} \AA^{-2})	ΔE_0 (eV)	R-factor (%)
Aged Pt ₁ (3)/CNT	2.5–11.2	1.2–3.4	Pt–N	4.0 (± 0.5)	2.02 (± 0.01)	4.12 (± 1.37)	14.37 (± 1.41)	1.3
			Pt–Cl	0.6 (± 0.1)	2.34 (± 0.02)	1.00*		
			Pt···C	4.1 (± 0.8)	2.99 (± 0.04)	10.64 (± 5.49)		

Pt–N indicates a single scattering path of the first-shell. Pt···C indicates a single scattering path of the second-shell (Shell column). The CN is the coordination number obtained from the amplitude reduction factor (S_0^2) of 0.84. R indicates bond distance. σ^2 indicates the Debye-Waller factor. ΔE_0 indicates the energy shift. R-factor was obtained from the best fit for the respective catalysts. (*Defined parameter to reduce correlations between variables)

Action: We revised the EXAFS fitting results of the aged Pt₁(3)/CNT in the revised Supplementary Information (Supplementary Fig. 12 and Supplementary Table 2).

5. Besides operando ICP-MS, operando Pt-L3 XANES and EXAFS as well as operando XPS of O 1s and Pt 4f should be performed to precisely identify catalyst operando behaviors.

Response: Following the reviewer's suggestion, we conducted *in situ* X-ray absorption spectroscopy (XAS) measurements to investigate the catalyst structure under CER operating conditions. The XAS results of Pt₁(X)/CNTs (X = 0.15, 1, and 3) were collected at open circuit potential (OCP) and 1.45 V_{RHE} in Ar-saturated 0.1 M HClO₄ and 0.1 M HClO₄ + 1 M NaCl electrolytes. Here, we would like to summarize the three main findings from the *in situ* XAS investigations:

A. *In situ* XANES spectra reveal the change of Pt oxidation state induced by the coverage of reaction intermediates on the Pt sites.

: The white line (WL) intensity at 11,689 eV represents the oxidation state of Pt. While the *ex situ* XANES spectrum of Pt₁(3)/CNT (powdery sample) reveals a WL intensity of a +2 oxidation state, the WL intensity marginally increases after immersing the catalyst into the electrolytes (Fig. R12a). At a potential of 1.45 V_{RHE}, the WL intensity further intensifies due to the adsorption of the CER intermediate, and this result agrees with our previous study [*ACS Catal.* **11**, 12232 (2021)]. Similarly, all Pt SACs exhibit increased WL intensities at 1.45 V_{RHE} compared to those at OCP. Interestingly, the WL intensity slightly increases in lower Pt contents, inferring the higher coverage of the CER intermediates as the proportion of Pt-N₃(V) increases. In the absence of NaCl, such WL changes are marginal. Therefore, the XANES spectra further corroborate our main finding that Pt-N₃(V) is a main catalytic site towards CER.

B. *In situ* EXAFS spectra reveal the formation of additional Pt-O bonds at the Pt-N₃(V) site under operating conditions.

: We observed an increase in CN of Pt-N bond for Pt₁(0.15)/CNT from 3 (Fig. 3b in the original manuscript) to 4 (Fig. R12b and Table R4) when the powdery sample/electrode was immersed in Ar-saturated 0.1 M HClO₄ + 1 M NaCl electrolyte at both OCP and 1.45 V_{RHE}. Since scattering paths of Pt-N and Pt-O are very similar and hardly distinguishable, we attribute this peak increment to a formation of additional Pt-O bond at the Pt-N₃(V) sites. This result supports our DFT prediction that the Pt-N₃(V) sites catalyze CER *via* the *OCl path.

C. *In situ* EXAFS spectra identify that the main catalytic site is Pt-N₃(V).

: *In situ* EXAFS spectra, measured at 1.45 V_{RHE} in an Ar-saturated 0.1 M HClO₄ + 1 M NaCl electrolyte, reveal that the Pt-Cl scattering peak newly emerges at 2.3 Å (Fig. R12b), which is not found at OCP or in NaCl-free electrolyte. Although we proposed *OCl formation at Pt-N₃(V), this peak infers that the OCl intermediate is vented towards the Pt site as a bridge form as predicted by DFT calculation (Fig. 4c,d in the original manuscript). Interestingly, the Pt-Cl scattering peak gradually increases as Pt content in Pt₁(X)/CNTs decreases, and the ratio between CNs of Pt-N/O and Pt-Cl converges to approximately 4 : 1 for Pt₁(0.15)/CNT (Table R4). This result indicates almost 100% utilization of the Pt site for catalyzing CER on Pt₁(0.15)/CNT, which solely consists of Pt-N₃(V) sites. As increasing Pt content, namely increasing the proportion of symmetric Pt-N₄ site, the value of CN_{Pt-N/O}/CN_{Pt-Cl} substantially increases, supporting again our conclusion that Pt-N₃(V) sites are the main catalytic sites in CER.

Fig. R12: **a** The Pt L₃-edge *in situ* XANES spectra of Pt_i(X)/CNTs (X = 0.15, 1, and 3) measured in Ar-saturated 0.1 M HClO₄ + 1 M NaCl electrolyte. The considerable increment in WL intensity was not found in NaCl-free electrolyte (not shown). **b** The k³-weighted Pt L₃-edge *in situ* EXAFS spectra of Pt_i(X)/CNTs (X = 0.15, 1, and 3) measured in Ar-saturated 0.1 M HClO₄ + 1 M NaCl electrolyte. The detailed EXAFS fitting parameters are provided in Table R4.

Table R4: Summary of EXAFS fitting parameters of Pt_i(X)/CNT (X = 0.15, 1, and 3) measured in Ar-saturated 0.1 M HClO₄ + 1 M NaCl electrolyte.

Sample	k range	R range	Shell	CN	R (Å)	σ ² (10 ⁻³ Å ⁻²)	Δ E ₀ (eV)	R -factor (%)
Pt _i (0.15)/CNT @ OCP	2.7–11.2	1.2–2.6	Pt–N/O	3.9 (± 0.7)	1.99 (± 0.01)	3.04 (± 1.48)	10.15 (± 2.21)	1.9
Pt _i (0.15)/CNT @ 1.45 V _{RHE}			Pt–N/O	4.1 (± 0.6)	2.00 (± 0.03)	2.50 (± 1.84)	12.74 (± 3.67)	2.4
			Pt–Cl	1.0 (± 0.3)	2.31 (± 0.03)	1.00*		
Pt _i (1)/CNT @ OCP			Pt–N/O	4.1 (± 0.6)	1.98 (± 0.01)	3.10 (± 1.28)	9.49 (± 2.84)	2.3
Pt _i (1)/CNT @ 1.45 V _{RHE}			Pt–N/O	4.2 (± 0.6)	1.99 (± 0.02)	2.44 (± 1.72)	9.32 (± 3.83)	2.3
			Pt–Cl	0.8 (± 0.3)	2.30 (± 0.04)	1.00*		
Pt _i (3)/CNT @ OCP			Pt–N/O	4.1 (± 0.7)	1.99 (± 0.01)	3.85 (± 1.50)	11.06 (± 2.12)	1.3
Pt _i (3)/CNT @ 1.45 V _{RHE}			Pt–N/O	4.2 (± 0.6)	2.03 (± 0.02)	3.54 (± 1.84)	12.65 (± 3.55)	2.2
			Pt–Cl	0.5 (± 0.3)	2.33 (± 0.04)	1.00*		

Pt–N/O indicates a single scattering path of the first-shell. Pt···C scattering path of the second-shell is excluded

to minimize correlations between multiple variables. The CN is the coordination number obtained from the amplitude reduction factor (S_0^2) of 0.84. R indicates bond distance. σ^2 indicates the Debye-Waller factor. ΔE_0 indicates the energy shift. R -factor was obtained from the best fit for the respective catalysts. (*Defined parameters to reduce correlations between variables)

We note that the *in situ* X-ray photoelectron spectroscopy (XPS) measurements have not been conducted due to technical issues related to the experimental conditions. Currently, we have developed an *in situ* XPS instrument at the Pohang Accelerator Laboratory (PAL) in collaboration with Dr. Ki-Jeong Kim, a beamline scientist at PAL. Unfortunately, however, the use of chlorine and its derivatives, such as Cl^- and ClO_4^- , is prohibited due to the PAL beamline regulations, which prevent the possible contamination of the XPS chamber by the Cl-derivatives and the possible risk of explosion caused by the accumulation of the perchlorate salts [*Sodium perchlorate; MSDS No. 1716, New Jersey Department of Health and Senior Service, Trenton, NJ, 2002*]. Since the electrolyte for CER contains chloride (NaCl) and perchlorate (HClO_4) salts, conducting an *in situ* XPS experiment is not feasible. Nevertheless, as the *in situ* XAS results clearly elucidate the local chemical structure of $\text{Pt}_1(\text{X})/\text{CNTs}$ under the CER operation conditions, we believe that our additional *in situ* XAS investigation sufficiently supports our primary findings.

Action: We newly discussed the experimental evidence of the formation of a reaction intermediate, *i.e.*, $^*\text{OCl}$, on the $\text{Pt}-\text{N}_3(\text{V})$ sites in the revised manuscript (copied below) and provided relevant *in situ* XAS results in the revised Supplementary Information (Supplementary Fig. 15 and Supplementary Table 3).

[Main manuscript/RESULTS] *In addition, the critical role of $\text{Pt}-\text{N}_3(\text{V})$ in catalyzing CER is further corroborated by *in situ* XAS measurements. In the Pt L₃-edge XANES spectra, which were measured in an Ar-saturated 0.1 M $\text{HClO}_4 + 1 \text{ M NaCl}$ electrolyte, the white line (WL) intensity marginally increases after immersing the catalyst into the electrolyte, and the increment further intensifies at 1.45 V_{RHE} (Supplementary Fig. 15). This result agrees well with our previous study and infers the adsorption of CER intermediates on the Pt sites³⁰. Interestingly, the intensified WL at 1.45 V_{RHE} is slightly higher for $\text{Pt}_1(0.15)/\text{CNT}$ and decreases with increasing Pt content in the catalysts, indicating higher coverage of CER intermediates as a proportion of $\text{Pt}-\text{N}_3(\text{V})$ sites in the catalysts increases. The same conclusion is also made with *in situ* EXAFS spectra that show an increasing $\text{Pt}-\text{Cl}$ scattering peak at 2.3 Å as Pt content in the catalysts decreases (Supplementary Fig. 15 and Supplementary Table 3). We further note that $\text{CN}_{\text{Pt}-\text{N}}$ of $\text{Pt}_1(0.15)/\text{CNT}$ increases from 3 for the powdery sample (Fig. 3c) to 4 under the electrochemical conditions and attribute this change to *in situ* formation of an additional $\text{Pt}-\text{O}$ bond, which will be discussed again in the DFT section. The ratio between $\text{CN}_{\text{Pt}-\text{N/O}}$ and $\text{CN}_{\text{Pt}-\text{Cl}}$ of $\text{Pt}_1(0.15)/\text{CNT}$ is approximately 4 : 1, and this supports the predominant presence of $\text{Pt}-\text{N}_3(\text{V})$ sites in $\text{Pt}_1(0.15)/\text{CNT}$ and their high catalytic activity towards CER.*

6. Necessary experiments and calculations should be conducted to support the claim of “This study suggests general guidelines for achieving high electrocatalytic performance of carbon-based SACs based on all other d8 metal ions, e.g., Ni^{II} and Pd^{II} as well as Pt^{II}”.

Response: In this study, we aimed to identify the chemical nature of the active site for Pt SACs. While there has been significant progress in high-performing Pt SACs, discrepancies between experimental and theoretical findings still persist. Experimentally, Pt SACs with Pt^{II} exhibit remarkable catalytic activity even in a square planar geometry (D_{4h} symmetry), while theoretical analyses predict its poor catalytic activity. The present work addresses this paradox by demonstrating that Pt^{II} sites with broken geometric symmetry are responsible for the exceptional catalytic performance of Pt SACs.

Considering that other transition metal SACs with d^8 electron configurations may also have a square planar geometry, we believed that the insights from Pt SACs could extend to the other d^8 metal ions, such as Ni^{II} and Pd^{II} [*Angew. Chem. Int. Ed.* **8**, 35 (1969)]. In fact, similar conundrums have arisen for the Ni SACs (and also Pd SACs), of which the active site is also theoretically predicted as a broken geometric symmetry. Some experimental investigations have also proposed the importance of Ni SAC sites with a broken symmetric

geometry for achieving better CO₂ electrolysis [*ACS Catal.* **10**, 10920 (2020); *J. Am. Chem. Soc.* **143**, 925 (2021)]. In light of these findings, our proposed guidelines might be applicable to other metal ions with a *ds* electronic configuration in SACs, not limited to Pt^{II} SACs, as discussed in ABSTRACT and RESULTS in the original manuscript. Nevertheless, we would like to moderate the overall tone of the relevant statements since it has yet to be verified, and further experiments with well-controlled model catalysts — which are beyond the scope of the present work — are still needed.

Action: In the revised manuscript, we moderated the overall tone of the relevant statements that our finding provides general guidelines for achieving the high electrocatalytic performance of carbon-based SACs based on all other *d*⁸ metal ions.

[Main manuscript/ABSTRACT] *This study may afford general guidelines for achieving a high electrocatalytic performance of carbon-based SACs based on other *d*⁸ metal ions.*

[Main manuscript/RESULTS] *Considering the cases of Ni^{II} SACs for electrochemical CO₂ reduction²⁰, these guidelines might be general tasks for other metal ions with a *d*⁸ electronic configuration in SACs, not limited to Pt^{II} catalytic sites.*

REVIEWERS' COMMENTS

Reviewer #1 (Remarks to the Author):

The revised version can be accepted.

Reviewer #2 (Remarks to the Author):

The authors have addressed all the issues about the computational details I raised. I, therefore, recommend it for publication.

Reviewer #3 (Remarks to the Author):

We are pleased with the authors' responses and the revised version. Therefore, it can be accepted now.